# Taming Momentum: Rethinking Optimizer States Through Low-Rank Approximation

**Zhengbo Wang**[1,2]  **Jian Liang**[2,3*]  **Ran He**[2,3]  **Zilei Wang**[1]  **Tieniu Tan**[2,3,4]

[1] University of Science and Technology of China
[2] NLPR & MAIS, Institute of Automation, Chinese Academy of Sciences
[3] School of Artificial Intelligence, University of Chinese Academy of Sciences
[4] Nanjing University
`zhengbowang@mail.ustc.edu.cn`, `liangjian92@gmail.com`

## Abstract

Modern optimizers like Adam and Muon are central to training large language models, but their reliance on first- and second-order momenta introduces significant memory overhead, which constrains scalability and computational efficiency. In this work, we reframe the exponential moving average (EMA) used in these momenta as the training of a linear regressor via online gradient flow. Building on this equivalence, we introduce LoRA-Pre, a novel low-rank optimizer designed for efficient pre-training. Specifically, LoRA-Pre reduces the optimizer's memory footprint by decomposing the full momentum matrix into a compact low-rank subspace within the online linear learner, thereby maintaining optimization performance while improving memory efficiency. We empirically validate LoRA-Pre's efficacy by pre-training models from the Llama architecture family, scaling from 60M to 1B parameters. LoRA-Pre achieves the highest performance across all model sizes. Notably, LoRA-Pre demonstrates remarkable rank efficiency, achieving comparable or superior results using only 1/8 the rank of baseline methods. Beyond pre-training, we evaluate LoRA-Pre's effectiveness in fine-tuning scenarios. With the same rank, LoRA-Pre consistently outperforms all efficient fine-tuning baselines. Specifically, compared to standard LoRA, LoRA-Pre achieves substantial improvements of 3.14 points on Llama-3.1-8B and 6.17 points on Llama-2-7B, validating our approach's effectiveness across both pre-training and fine-tuning paradigms. Our code is publicly available at https://github.com/mrflogs/LoRA-Pre.

## 1 Introduction

Large Language Models (LLMs) (Guo et al., 2025; Yang et al., 2025; Grattafiori et al., 2024; Brown et al., 2020; Comanici et al., 2025; Touvron et al., 2023; Jaech et al., 2024) have become the centerpiece of modern deep learning. Trained on trillions of tokens from heterogeneous sources and scaled to unprecedented parameter counts, they demonstrate remarkable generalization and transfer capabilities. Beyond this, some LLMs have reasoning abilities and leverage external tools (Guo et al., 2025; Yang et al., 2025). These advances have transformed LLMs from statistical learners into versatile systems, driving breakthroughs across research and real-world applications.

However, the success of LLMs comes with formidable training and adaptation costs (Grattafiori et al., 2024). The vast number of trainable parameters demands enormous memory and computational resources during pre-training and fine-tuning. A key contributor to this burden lies in the optimizer states. For instance, Adam (Kingma & Ba, 2015), the *de facto* optimizer for training LLMs, maintains not only the model weights but also first- and second-order moment estimates of the gradients. This triples memory usage and further exacerbates scalability bottlenecks, underscoring the urgent need for more efficient optimization strategies.

---

[*] Corresponding author.

To address this, a series of low-rank optimization methods has emerged. One prominent line of research achieves optimizer state compression through projected gradient descent (Zhao et al., 2024; Chen et al., 2024; Hao et al., 2024; Modoranu et al., 2025). These methods initialize projection matrices via SVD or random mappings, project gradients into smaller subspaces for optimizer state computation, and then project them back to achieve parameter updates, thereby compressing the optimization overhead. Additionally, such methods require periodic subspace updates to enable high-rank parameter updates following $W = \Delta W_{T_1} + \Delta W_{T_2} + \cdots$. However, due to the inability to update subspaces instantly, error accumulation occurs in optimizer state computation, leading to suboptimal performance. This motivates the need for a more dynamic approach that can rapidly adapt to changing gradient subspaces.

In this paper, we propose LoRA-Pre, a novel low-rank optimizer for LLM pre-training that addresses these limitations through a different approach. Our key insight is an interesting mathematical connection between the exponential moving average (EMA) of momentum and linear regression. Specifically, we demonstrate that EMA momentum updates are mathematically equivalent to training an online linear regressor with gradient descent on the online gradient flow:

$$m \leftarrow \beta \cdot m + (1 - \beta) \cdot g \quad \Longleftrightarrow \quad \min_m L(m; g) = \frac{1}{2} \cdot \|m - g\|_F^2, \tag{1}$$

where $m \in \mathbb{R}^{p \times q}$ represents the momentum, $g$ is the online gradient, and $\beta$ is the momentum coefficient. This equivalence reveals that momentum accumulation can be viewed as fitting a linear model to approximate the gradient history.

Leveraging this theoretical insight, we develop a memory-efficient optimizer through momentum compression via low-rank factorization. Instead of maintaining the full momentum matrix $m$, we decompose it as the product of two low-rank matrices: $m = m_B \cdot m_A$, where $m_B \in \mathbb{R}^{p \times r}$ and $m_A \in \mathbb{R}^{r \times q}$, with $r \ll \min(p, q)$. This factorization reduces memory complexity from $p \times q$ to $(p + q) \times r$, yielding substantial memory savings for large-scale models. The low-rank momentum is then updated by solving $\min_{m_B, m_A} L(m_B, m_A; g) = \frac{1}{2} \cdot \|m_B m_A - g\|_F^2$, with explicit update rules derived in Theorem 3.1.

This theoretical framework enables us to compress any momentum-based optimizer. We demonstrate its versatility by developing LoRA-Pre variants for both Adam (Kingma & Ba, 2015) and Muon (Jordan et al., 2024) optimizers, with detailed algorithms provided in Appendix B. Extensive experiments across pre-training and fine-tuning tasks validate the effectiveness of our method, while ablation studies demonstrate strong robustness across different rank variations.

Our main contributions are summarized as follows:

- We establish a novel theoretical connection showing that exponential moving average (EMA) momentum updates are mathematically equivalent to training a linear regressor via online gradient flow.

- Based on this insight, we propose LoRA-Pre, a memory-efficient low-rank optimizer for pre-training that compresses optimizer states by factorizing the momentum matrix into low-rank components. We construct LoRA-Pre variants for both Adam and Muon optimizers, mathematically derive their low-rank update rules through our regression formulation, and achieve substantial memory reduction while preserving optimization dynamics.

- We provide extensive experimental validation across both pre-training and fine-tuning tasks, demonstrating that LoRA-Pre achieves superior performance with remarkable rank efficiency compared to existing baselines, confirming both the efficiency and effectiveness of our approach across diverse model scales and application scenarios.

## 2 RELATED WORKS

**Low-Rank Adaptation.** The scaling of Large Language Models (LLMs) has spurred the development of Parameter-Efficient Fine-Tuning (PEFT) methods (Hu et al., 2022; Liu et al., 2024; Wang et al., 2025; Ding et al., 2023; Liu et al., 2022; 2023; Hayou et al., 2024; Wang et al., 2024a; Edalati et al., 2023; Zhang et al., 2023; Tastan et al., 2025; Wang et al., 2023; 2024c;b), which aim to adapt pre-trained models to downstream tasks with reduced computational and memory overhead. Among

these PEFT methods, Low-Rank Adaptation (LoRA) (Hu et al., 2022) and its variants (Wang et al., 2025; 2024a; Hayou et al., 2024; Liu et al., 2024; Yen et al., 2025) have emerged as the predominant methods in the field.

LoRA is grounded in the principle that weight updates during fine-tuning possess an intrinsic low-rank structure (Aghajanyan et al., 2021). By re-parameterizing these updates as the product of two low-rank matrices, LoRA substantially reduces the number of trainable parameters while maintaining competitive performance, thereby enabling efficient adaptation of LLMs with limited computational resources. The effectiveness of LoRA has inspired a line of research aimed at addressing its shortcomings. For instance, LoRA+ (Hayou et al., 2024) introduces differential learning rates for the two low-rank matrices to improve convergence and final task performance. DoRA (Liu et al., 2024) decomposes pre-trained weights into magnitude and direction components, applying LoRA specifically to the directional component to better capture fine-tuning dynamics. Recent works like LoFT (Tastan et al., 2025) and LoRA-Pro (Wang et al., 2025) establish theoretical connections between LoRA and full fine-tuning via projected gradient equivalents.

While effective for fine-tuning, existing LoRA-based methods face fundamental challenges when applied to pre-training from scratch. Unlike fine-tuning, where small adaptations naturally exhibit low-rank structure, pre-training from random initialization requires full-rank weight updates to learn diverse representations across the entire parameter space (Lialin et al., 2024; Kamalakara et al., 2022). This mismatch between LoRA's low-rank assumption and pre-training's full-rank requirements results in suboptimal performance in the pre-training stage.

**Low-Rank Pre-Training.** The pre-training cost of LLMs has surged dramatically with the rapid expansion of model scale. A promising direction for mitigating these costs is compressing optimizer states into a low-rank subspace, a strategy that significantly reduces memory footprint and communication overhead (Zhao et al., 2024; Modoranu et al., 2025; Ma et al., 2025; Han et al., 2024; Zmushko et al., 2025; Chen et al., 2024; Hao et al., 2024; Shen et al., 2025; Mahdavinia & Mahdavi, 2025; Zhang et al., 2025). For instance, GaLore (Zhao et al., 2024) utilizes Singular Value Decomposition (SVD) to project gradient information into a low-rank subspace for state compression, subsequently projecting the optimized gradients back for parameter updates. To enhance computational efficiency, Flora (Hao et al., 2024) substitutes the expensive SVD operation with random projection, while Fira (Chen et al., 2024) incorporates SGD momentum to leverage gradient information from the complementary subspace. However, these projection-based methods typically rely on *periodic* subspace updates to amortize costs, which often results in optimization discontinuities and error accumulation due to the lag in subspace adaptation.

Recent works have explored online strategies to address these limitations. MLorc (Shen et al., 2025) employs randomized SVD for online momentum compression. MoFaSGD (Mahdavinia & Mahdavi, 2025) utilizes momentum factorization to approximate full-rank momentum online, ensuring non-convex convergence. Similarly, ADAPM (Zhang et al., 2025) compresses first-order momentum into a low-rank subspace via linear regression. In contrast, our proposed LoRA-Pre fundamentally reformulates momentum maintenance as an online regression task. By directly evolving low-rank factors via online gradient flow at every step, our approach achieves continuous subspace adaptation, effectively eliminating the instabilities associated with periodic updates or heuristic approximations.

## 3 METHOD

We begin by revisiting the *de facto* standard optimizer, Adam (Kingma & Ba, 2015), in Section 3.1. Then, we establish a connection between the exponential moving average and an online linear regressor over past gradients in Section 3.2. Finally, Section 3.3 introduces our efficient optimizer, LoRA-Pre, which compresses optimizer states through low-rank parameterization.

### 3.1 PRELIMINARY

We first review Adam (Kingma & Ba, 2015), the *de facto* optimizer in modern deep learning, which combines the benefits of AdaGrad (Duchi et al., 2011) and RMSProp (Hinton et al., 2012) by maintaining estimates of the first and second moments of gradients to achieve adaptive learning rates and robust performance.

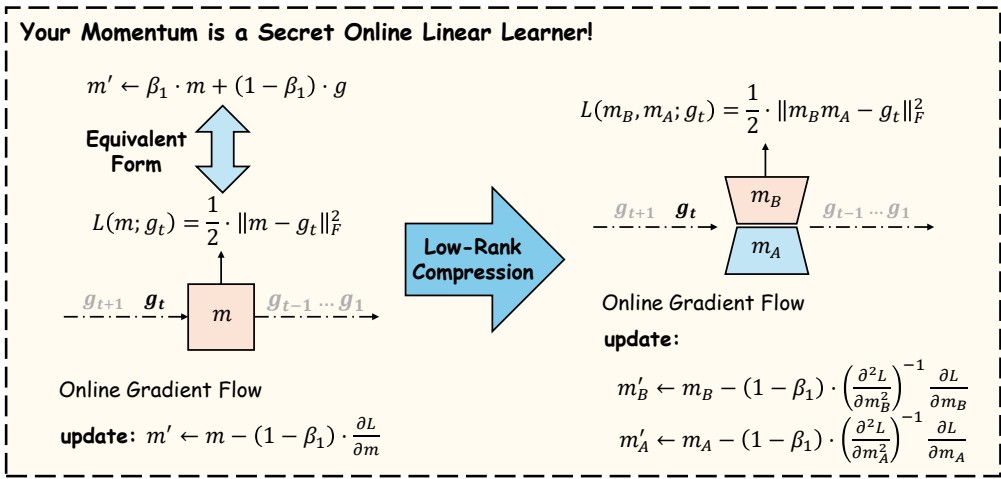

Figure 1: **Illustration of our LoRA-Pre method.** In this work, we establish a novel connection: *the exponential moving average (EMA) update for optimizer momentum is mathematically equivalent to training a linear regressor using online gradient descent.* Leveraging this equivalence, we propose compressing the optimizer states (i.e., the momenta) using low-rank matrices to reduce the memory footprint. Finally, the closed-form update rules for these matrices without requiring backpropagation are given by Theorem 3.1.

Consider an optimization problem where $x_t \in \mathcal{X}$ represents a data point drawn from a distribution $p_{data}$, $\mathcal{L}(\cdot) : \mathcal{X} \to \mathbb{R}$ is a loss function, and $\theta \in \mathbb{R}^d$ are the optimized parameters. The Adam optimizer (Kingma & Ba, 2015) updates $\theta$ according to the following steps:

$$g_t = \frac{\partial \mathcal{L}(x_t)}{\partial \theta}, \quad x_t \sim p_{data}(x), \qquad \text{(Gradient Computation)} \qquad (2)$$

$$m_t = \beta_1 \cdot m_{t-1} + (1 - \beta_1) \cdot g_t, \qquad \text{(EMA of the First Moment)} \qquad (3)$$

$$v_t = \beta_2 \cdot v_{t-1} + (1 - \beta_2) \cdot g_t^2, \qquad \text{(EMA of the Second Moment)} \qquad (4)$$

$$\hat{m}_t = \frac{m_t}{1 - \beta_1^t}, \quad \hat{v}_t = \frac{v_t}{1 - \beta_2^t}, \qquad \text{(Bias-Correction)} \qquad (5)$$

$$\theta_{t+1} = \theta_t - \frac{\gamma}{\sqrt{\hat{v}_t} + \epsilon} \cdot \hat{m}_t. \qquad \text{(Parameter Update)} \qquad (6)$$

Here, $m_t$ and $v_t$ represent the Exponential Moving Average (EMA) of the first- and second-order moments, respectively. The hyper-parameters include the learning rate $\gamma$, the exponential decay rates $\beta_1, \beta_2 \in [0, 1)$ for the moment estimates, and a small constant $\epsilon > 0$ for numerical stability.

Similar to the Adam optimization process, momentum also plays a critical role in other modern optimizers (Shazeer & Stern, 2018; Jordan et al., 2024), enhancing stability and convergence. However, storing momentum states introduces significant memory overhead. Our work directly addresses this by compressing the momentum term to reduce the optimizer's memory footprint.

### 3.2 YOUR MOMENTUM IS A SECRET ONLINE LINEAR REGRESSOR

To begin with, we reveal an interesting connection: momentum updates in modern optimizers are secretly performing online linear regression. Specifically, updating the momentum $m$ via EMA is mathematically equivalent to optimizing $m$ as the parameter of a linear regressor using online gradient flow.

To illustrate this, consider the first-order momentum as an example. The standard EMA update for the first-order momentum can be rewritten as follows:

$$m_{t+1} = \beta \cdot m_t + (1 - \beta) \cdot g, \qquad (7)$$

$$= \underbrace{m_t}_{weight} - \underbrace{(1 - \beta)}_{lr} \cdot \underbrace{(m_t - g)}_{gradient}. \qquad (8)$$

As shown in Equation (8), the EMA update is mathematically equivalent to a gradient descent step where the parameter being optimized is the momentum $m$, the learning rate is $1 - \beta$, and the gradient is $\frac{\partial L(m_t; g)}{\partial m} = m_t - g$. This reformulation reveals that EMA updates essentially function as an online linear regressor that continuously adjusts the momentum weights based on incoming gradients. The underlying objective being minimized is:

$$\min_m L(m; g) = \frac{1}{2} \cdot \|m - g\|_F^2. \tag{9}$$

This insight opens a new avenue for optimizer footprint optimization: since momentum parameters are linear model weights, we can apply standard model compression techniques to reduce optimizer memory usage during training.

### 3.3 LoRA-Pre: Low-Rank Online Linear Regression

We now introduce LoRA-Pre, a new low-rank optimizer for pre-training. Building on the equivalence between exponential moving averages and online linear regression, LoRA-Pre compresses the momentum term $m$ via a low-rank factorization, inspired by the LoRA technique (Hu et al., 2022). This approach can be applied to any momentum-based optimizer, such as Adam (Kingma & Ba, 2015) and Muon (Jordan et al., 2024). We detail the compression strategies for both first- and second-order momentum terms below.

**First-Order Momentum Compression.** Having established that momentum updates are equivalent to gradient descent on the objective $\min_m L(m; g) = \frac{1}{2} \cdot \|m - g\|_F^2$ in Section 3.2, we can now apply low-rank compression to reduce memory usage. Instead of storing and updating the full momentum matrix $m \in \mathbb{R}^{p \times q}$ directly, we decompose it into the product of two low-rank matrices $m_B \in \mathbb{R}^{p \times r}$ and $m_A \in \mathbb{R}^{r \times q}$, $r \ll \min(p, q)$, i.e., $m = m_B \cdot m_A$. This factorization transforms our original optimization problem into:

$$\min_{m_B, m_A} L(m_B, m_A; g) = \frac{1}{2} \cdot \|m_B m_A - g\|_F^2. \tag{10}$$

To maintain memory efficiency, we solve this optimization problem using standard gradient descent on the factorized matrices $m_B$ and $m_A$. To ensure computational efficiency, we derive closed-form update rules for these matrices without requiring back-propagation, which is given by Theorem 3.1. We resort to Newton's method for updating since the solution can be expressed in the form of EMA.

> **Theorem 3.1.** *Assume both matrices $m_B \in \mathbb{R}^{p \times r}$ and $m_A \in \mathbb{R}^{r \times q}$ are full rank. For the objective $\min_{m_B, m_A} L(m_B, m_A; g) = \frac{1}{2} \cdot \|m_B m_A - g\|_F^2$, Newton's method yields the following closed-form update rules:*
>
> $$m_B \leftarrow (1 - \gamma_1) \cdot m_B + \gamma_1 \cdot g m_A{}^T (m_A m_A^T)^{-1}, \tag{11}$$
> $$m_A \leftarrow (1 - \gamma_1) \cdot m_A + \gamma_1 \cdot (m_B^T m_B)^{-1} m_B^T g. \tag{12}$$
>
> *Here, $\gamma_1$ is the learning rate for the factorized optimization problem.*
>
> *Proof.* See Appendix A. $\square$

**Second-Order Momentum Compression.** The compression of second-order momentum $v$ presents additional challenges due to the constraints imposed by Adam's parameter update rule. Since Equation (6) requires the square root of momentum, i.e., $\sqrt{v}$, the second-order momentum must be element-wise positive.

A naive approach would parameterize the second momentum as $v = v_B \cdot v_A$ and optimize using the regression loss $L(v_B, v_A; g) = \frac{1}{2} \cdot \|v_B v_A - g^2\|_F^2$. From Theorem 3.1, we derive the corresponding parameter update rule:

$$v_B \leftarrow (1 - \gamma_2) \cdot v_B + \gamma_2 \cdot g^2 v_A^T (v_A v_A^T)^{-1}, \tag{13}$$
$$v_A \leftarrow (1 - \gamma_2) \cdot v_A + \gamma_2 \cdot (v_B^T v_B)^{-1} v_B^T g^2. \tag{14}$$

Unfortunately, this approach cannot guarantee that $v_{i,j} > 0, \forall i, j$, making the computation of $\sqrt{v} = \sqrt{v_B v_A}$ problematic.

To address this issue, we re-parameterize the second-order momentum as $v = (v_B \, v_A)^{\circ 2}$, where $\circ$ denotes the Hadamard product. This re-parameterization ensures element-wise positivity while maintaining the low-rank structure. We then formulate the optimization of low-rank matrices $v_B$ and $v_A$ as:

$$\min_{v^B, v^A} L(v_B, v_A; g) = \frac{1}{2} \cdot \|v_B v_A - |g|\|_F^2. \tag{15}$$

Its update rule can thus be directly derived from Theorem 3.1.

$$v_B \leftarrow (1 - \gamma_2) \cdot v_B + \gamma_2 \cdot |g| v_A^T (v_A v_A^T)^{-1}, \tag{16}$$

$$v_A \leftarrow (1 - \gamma_2) \cdot v_A + \gamma_2 \cdot (v_B^T v_B)^{-1} v_B^T |g|. \tag{17}$$

**Low-Rank Optimizer Algorithms.** As shown before, our method can be applied to any optimizer with momentum to compress its optimizer state during pre-training and fine-tuning stages. The detailed pseudo-codes of LoRA-Pre optimizer for AdamW (Kingma & Ba, 2015) and Muon (Jordan et al., 2024) are provided in Appendix B.

## 4 EXPERIMENTAL RESULTS

In this section, we present extensive experiments to evaluate the effectiveness of our proposed method, LoRA-Pre. Our evaluation encompasses both memory-efficient pre-training and memory-efficient fine-tuning on downstream tasks.

We begin by assessing LoRA-Pre's pre-training capabilities in Section 4.1. Following the experimental setup of GaLore (Zhao et al., 2024), we train Llama (Touvron et al., 2023) models from scratch with varying model sizes of 60M, 130M, 350M, and 1B parameters. All models are trained on the Colossal Clean Crawled Corpus (C4) dataset (Raffel et al., 2020), a large-scale cleaned dataset specifically designed for language model pre-training. To simulate realistic pre-training conditions, the models are trained on sufficiently large volumes of data without repetition.

Subsequently, we evaluate LoRA-Pre's fine-tuning performance in Section **??**. We fine-tune both Llama-3.1-8B (Grattafiori et al., 2024) and Llama-2-7B (Touvron et al., 2023) models on a 100k subset sampled from the MetaMathQA dataset (Yu et al., 2024). The fine-tuned models are then evaluated on the GSM8K (Cobbe et al., 2021) and MATH-500 (Lightman et al., 2024) datasets. Finally, we present an ablation study of LoRA-Pre in Appendix 4.3.

**Implementation Details.** To ensure a fair comparison, we align the experimental setup with that of GaLore (Zhao et al., 2024). By default, LoRA-Pre is applied to all parameters in the attention and MLP layers, while other parameters are optimized using the standard Adam (Kingma & Ba, 2015) optimizer. We set the default ranks for the 60M, 130M, 350M, and 1B parameter models to 128, 256, 256, and 512, respectively. The optimal learning rate is selected from the set $\{0.01, 0.005, 0.001, 0.0005, 0.0001\}$ based on validation perplexity. To maintain strict fairness in comparison, we retain the same scale factor of 0.25 as used in GaLore (Zhao et al., 2024). For memory-efficient fine-tuning tasks, we set the default rank as 8 and set the learning rate as $2e - 5$ by default.

### 4.1 MEMORY-EFFICIENT PRE-TRAINING

In this section, we evaluate the pre-training performance of our proposed method, LoRA-Pre. Our experimental setup strictly follows that of GaLore (Zhao et al., 2024). We compare LoRA-Pre against several baseline methods, including both full optimizers and low-rank optimizers: 1) **Adam** (Kingma & Ba, 2015): The *de facto* optimizer in modern deep learning that utilizes first- and second-order momentum statistics to dynamically adjust learning rates and stabilize training. 2) **Muon** (Jordan et al., 2024): A novel preconditioned optimizer that updates parameters by orthogonalizing the first-order momentum. 3) **GaLore** (Zhao et al., 2024): A low-rank optimizer that projects gradients using SVD and computes optimizer states in a reduced subspace. 4) **Low-Rank** (Kamalakara et al., 2022): A traditional low-rank approach that directly represents weights through learnable low-rank factorization $W = BA$. 5) **LoRA** (Hu et al., 2022): The most widely adopted low-rank method for fine-tuning that factorizes weights as $W = W_0 + BA$. For pre-training scenarios, we maintain $W_0$ as the full-rank initialization matrix. 6) **ReLoRA** (Lialin et al., 2024): A LoRA variant designed for pre-training that periodically merges $BA$ into $W$ and initializes $BA$

Table 1: Comparison with low-rank algorithms on pre-training various sizes of Llama models on the C4 dataset. We report the validation perplexity (↓) on a hold-out C4 test set. The best and second-best performance within the low-rank optimizers are highlighted with **bold** and underline. ∗ denotes the results are reproduced by ourselves.

| Model Size | 60M | 130M | 350M | 1B |
|---|---|---|---|---|
| $r/d_{\text{model}}$ | 128 / 512 | 256 / 768 | 256 / 1024 | 512 / 2048 |
| Training Tokens | 1.1B | 2.2B | 6.4B | 13.1B |
| Adam (Kingma & Ba, 2015) | 34.09 | 25.08 | 18.80 | 15.56 |
| Muon (Jordan et al., 2024) | 28.43 | 21.86 | 16.17 | 13.41 |
| GaLore (Zhao et al., 2024) | 34.88 | 25.36 | 18.95 | 15.64 |
| Low-Rank (Kamalakara et al., 2022) | 78.18 | 45.51 | 37.41 | 142.53 |
| LoRA (Hu et al., 2022) | 34.99 | 33.92 | 25.58 | 19.21 |
| ReLoRA (Lialin et al., 2024) | 37.04 | 29.37 | 29.08 | 18.33 |
| SLTrain (Han et al., 2024) | 34.15 | 26.04 | 19.42 | 16.14 |
| LORO (Mo et al., 2025) | 33.96 | 24.59 | 18.84 | 15.19 |
| Fira* (Chen et al., 2024) | 31.19* | 24.51* | 17.22* | 14.31 |
| **LoRA-Pre Adam** | 32.57 | 23.78 | **16.36** | **13.53** |
| **LoRA-Pre Muon** | **30.76** | **23.05** | 16.97 | 13.92 |

with optimizer state resets. 7) **SLTrain** (Han et al., 2024): A sparse plus low-rank approach that parameterizes weights as $W = S + BA$, where both components are jointly optimized. 8) **LORO** (Mo et al., 2025): A method that optimizes LoRA parameters by strictly constraining updates within the low-rank manifold. 9) **Fira** (Chen et al., 2024): A method that improves upon GaLore with the Norm-Based Scaling and the Norm-Growth Limiter.

We pre-trained Llama series models of different sizes to evaluate LoRA-Pre against these baseline methods. By default, all low-rank optimizers are built upon the Adam (Kingma & Ba, 2015) optimizer foundation. To demonstrate the generalizability of our approach, we also evaluate LoRA-Pre with Muon (Algorithm 2), as our method is compatible with any momentum-based optimizer.

The results, presented in Table 1, demonstrate that our method achieves superior performance across multiple model scales. Specifically, LoRA-Pre Adam and LoRA-Pre Muon attain either the highest or second-highest performance across almost all four model sizes (60M, 130M, 350M, and 1B), validating the effectiveness of our approach. While Fira yields competitive results on the 60M model, LoRA-Pre consistently outperforms it on the larger scales (130M, 350M, and 1B), likely because our method avoids the error accumulation associated with Fira's projected gradients. Furthermore, LoRA-Pre Adam outperforms the previous best efficient baselines by substantial margins of 0.81, 2.45, and 1.6 perplexity points for the 130M, 350M, and 1B models, respectively. Furthermore, when integrated with the Muon (Jordan et al., 2024) optimizer, LoRA-Pre Muon achieves additional improvements on both 60M and 130M scale models, demonstrating our method's ability to generalize across different optimizers.

## 4.2 MEMORY-EFFICIENT FINE-TUNING

In this section, we evaluate the fine-tuning performance of LoRA-Pre on mathematical tasks. We fine-tune Llama-2-7B and Llama-3.1-8B models on the MetaMath100k dataset and assess their performance on GSM8K (Cobbe et al., 2021) and MATH-500 (Lightman et al., 2024). To ensure a fair comparison, we maintain identical hyper-parameters and training configurations across all methods.

We select several memory-efficient fine-tuning baselines for comparison, including: 1) **LoRA** (Hu et al., 2022): the standard low-rank fine-tuning method. 2) **rsLoRA** (Kalajdzievski, 2023): an improved LoRA variant that optimizes the scaling factor through rank-stabilized normalization. 3) **DoRA** (Liu et al., 2024): a LoRA extension that decomposes weight updates into magnitude and directional components for more effective optimization. 4) **GaLore** (Zhao et al., 2024): a memory-efficient optimizer that projects gradients into low-rank subspaces using SVD decomposition. To demonstrate cross-optimizer compatibility, we evaluate Muon-based versions, including: 1) **Ga-**

Table 2: **Results of memory-efficient fine-tuning methods.** We compare our method with efficient fine-tuning methods including LoRA (Hu et al., 2022), rsLoRA (Kalajdzievski, 2023), and DoRA (Liu et al., 2024), and an efficient optimizer GaLore (Zhao et al., 2024). The models are fine-tuned on the MetaMath100k (Yu et al., 2024) dataset, and evaluated on GSM8K (Cobbe et al., 2021) and MATH-500 (Lightman et al., 2024). We highlight the best performance on Adam-like optimizer and Muon-like optimizer with **bold**.

| Method | Llama-3.1-8B | | | Llama-2-7B | | |
|---|---|---|---|---|---|---|
| | GSM8K | MATH-500 | Average | GSM8K | MATH-500 | Average |
| LoRA (Hu et al., 2022) | 70.76 | 17.06 | 43.91 | 44.62 | 7.34 | 25.98 |
| rsLoRA (Kalajdzievski, 2023) | 71.06 | 17.46 | 44.26 | 48.79 | 5.75 | 27.27 |
| DoRA (Liu et al., 2024) | 71.06 | 17.86 | 44.46 | 44.39 | 6.55 | 25.47 |
| GaLore (Zhao et al., 2024) | 65.08 | **18.65** | 41.87 | 36.44 | **8.33** | 22.39 |
| **LoRA-Pre Adam** | **76.44** | 17.66 | **47.05** | **57.35** | 6.94 | **32.15** |
| GaLore Muon (Zhao et al., 2024) | 63.41 | 18.06 | 40.74 | 33.11 | 4.37 | 18.74 |
| LoRA Muon (Hu et al., 2022) | 70.30 | 19.25 | 44.78 | 35.15 | **6.15** | 20.65 |
| **LoRA-Pre Muon** | **72.65** | **20.83** | **46.74** | **47.20** | **6.15** | **26.68** |

**Lore Muon**: which apply the GaLore (Zhao et al., 2024) algorithm to the Muon optimizer, and 2) **LoRA Muon**: which optimizes LoRA with the Muon optimizer.

The results are presented in Table 2. LoRA-Pre consistently achieves the highest scores across all experimental configurations, demonstrating superior performance regardless of the base model or optimizer used. The improvements are particularly notable across different settings: when training Llama-3.1-8B with Adam, LoRA-Pre shows an average improvement of 2.59 points over the second-best method, while with Llama-2-7B and Adam, this improvement increases to 4.88 points. When using the Muon optimizer, LoRA-Pre maintains its advantage with improvements of 1.96 and 6.03 points for the respective models. These results confirm LoRA-Pre's effectiveness across diverse experimental conditions and its robust compatibility with different optimizers.

## 4.3 ABLATION STUDY

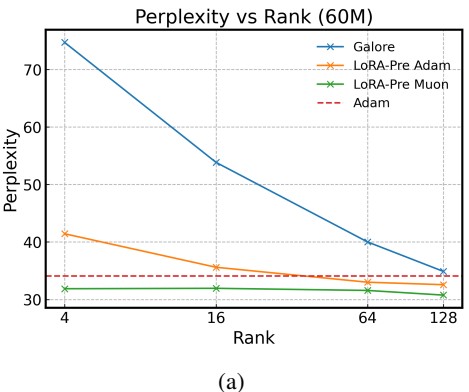
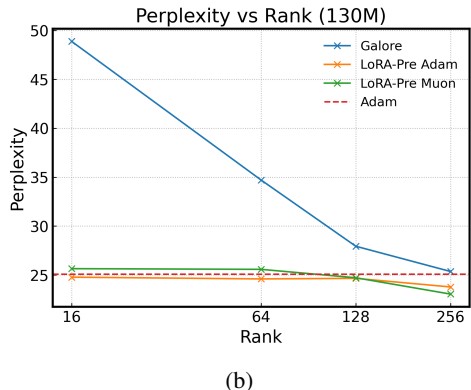

(a)  (b)

Figure 2: **Rank efficiency comparison across efficient optimization methods.** Perplexity versus rank for 60M (left) and 130M (right) models, demonstrating LoRA-Pre's superior performance at lower ranks compared to baseline methods.

**Ablation of Different Rank.** To systematically evaluate how rank selection affects the performance of LoRA-Pre compared to other efficient optimization methods, we conduct comprehensive experiments across different rank configurations. We evaluate LoRA-Pre (both Adam and Muon variants) against GaLore (Zhao et al., 2024) on 60M and 130M parameter models. We test ranks of {4, 16, 64, 128} for the 60M model and {16, 64, 128, 256} for the 130M model to observe performance trends across different memory budgets.

Figure 2 shows that all methods improve with increasing rank, but exhibit different rank efficiency. LoRA-Pre consistently achieves better perplexity at lower ranks compared to GaLore. First, all methods show improved performance with increasing rank, but they differ significantly in their rank efficiency. When comparing specific configurations, the efficiency differences become clear. On the 60M model, LoRA-Pre Adam at rank=16 achieves comparable performance to GaLore at rank=128, representing an $8\times$ reduction in rank requirement. Similarly, on the 130M model, LoRA-Pre Adam at rank=16 matches GaLore's performance at rank=256, representing a $16\times$ efficiency improvement. LoRA-Pre Muon shows higher rank tolerance than LoRA-Pre Adam. We attribute LoRA-Pre's rank efficiency to its continuous subspace adaptation mechanism. GaLore performs periodic subspace updates, creating intervals where the subspace becomes misaligned with the gradient structure. To compensate for this error accumulation, GaLore requires larger subspaces. In contrast, LoRA-Pre adjusts its subspace at each step, maintaining better alignment and thus achieving effective optimization with smaller subspaces.

To gain deeper insights into this rank efficiency, we examine the training dynamics of LoRA-Pre Muon across different rank configurations. Figure 3 visualizes the perplexity trajectories for the 130M model with ranks of 256, 128, 64, and 16.

The results reveal an intriguing convergence pattern: while smaller ranks initially exhibit higher perplexity values, this performance gap diminishes rapidly as training progresses. This behavior demonstrates that LoRA-Pre's dynamic subspace update mechanism can efficiently capture the evolving momentum structure during training, even when operating with constrained ranks. This rapid adaptation capability explains why LoRA-Pre maintains competitive performance across a wide range of rank settings, making it both robust to rank selection and practically appealing for memory-constrained training scenarios.

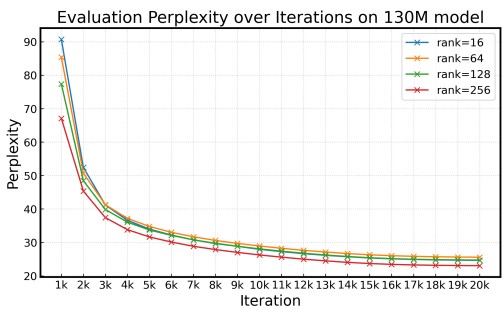

Figure 3: Test perplexity for LoRA-Pre Muon with different ranks during training.

Table 3: **Results of pre-training using different efficient Muon optimizers.**

| Model Size | 60M | 130M | 350M |
|---|---|---|---|
| Muon (Jordan et al., 2024) | 28.43 | 21.86 | 16.17 |
| Muon w/o momentum | 32.15 | 24.23 | 17.33 |
| GaLore Muon (Zhao et al., 2024) | 34.39 | 25.16 | 19.24 |
| Fira Muon (Chen et al., 2024) | 34.45 | 24.85 | 17.40 |
| **LoRA-Pre Muon** | **30.76** | **23.05** | **16.97** |

**Ablation of Low-Rank Muon Optimizers.** In this section, we evaluate the effectiveness of current efficient optimizers by extending them to the recently proposed Muon optimizer (Jordan et al., 2024). Since existing efficient optimizers were originally designed for Adam (Kingma & Ba, 2015), their compatibility and performance with other optimizers remain unexplored. We conduct experiments on 60M, 130M, and 350M parameter models, comparing LoRA-Pre against GaLore (Zhao et al., 2024) and Fira (Chen et al., 2024) by adapting their implementations to use Muon. Standard Muon serves as the upper bound, while Muon without momentum provides the lower bound. The Muon-based algorithm for LoRA-Pre is presented in Algorithm 2.

The results in Table 3 reveal two significant findings. First, LoRA-Pre Muon consistently outperforms all other efficient optimizers, achieving improvements of 3.54, 1.80, and 0.43 points over the second-best method at 60M, 130M, and 350M parameters, respectively. Second, projection-based methods surprisingly perform worse than basic Muon without momentum, despite incorporating momentum computation. This counterintuitive result exposes fundamental generalization limitations of projection-based gradient descent methods when applied to different optimizers. We attribute this phenomenon to the periodic subspace updates in projection-based methods, which introduce momentum computation errors that subsequently affect Muon's orthogonal update calculations. In

contrast, LoRA-Pre continuously updates its subspace, enabling better capture of the orthogonal space during Muon's update process and achieving superior performance.

## 5 CONCLUSION

In this paper, we present LoRA-Pre, a novel low-rank efficient optimizer. We establish that EMA momentum updates are mathematically equivalent to training an online linear regressor with gradient descent on the online gradient flow. Building on this insight, we propose compressing the momentum component through low-rank factorization, deriving update rules that maintain the EMA form while operating in a compressed parameter space. We provide two variants: LoRA-Pre Adam and LoRA-Pre Muon. Extensive experiments on pre-training and fine-tuning tasks demonstrate that LoRA-Pre achieves competitive or superior performance across all evaluated tasks and model sizes. Notably, our method exhibits excellent rank robustness, requiring only 1/8 or fewer ranks compared to previous methods while achieving comparable results. The approach generalizes effectively to various optimizers, making it a versatile solution for memory-efficient optimization.

## ACKNOWLEDGEMENT

This work was funded by the National Natural Science Foundation of China under Grants (62276256, U2441251, 62550062, 62425606, 32341009) and the Young Elite Scientists Sponsorship Program by CAST (2023QNRC001). We gratefully acknowledge Jie Cheng and Zhen Yang for providing computational resources and for their valuable discussions, which were critical to this work.

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

# Taming Momentum: Rethinking Optimizer States Through Low-Rank Approximation

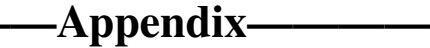

————Appendix————

The structure of the Appendix is as follows,

- Appendix A contains the proofs of the theorems in the main manuscript.
- Appendix B details the optimization algorithms of the proposed method.
- Appendix C provides theoretical analysis of approximation error and convergence of the proposed method.
- Appendix D provides additional experiments of our method.
- Appendix E details the LLM usage in this paper.

## A  PROOF OF THEORETICAL RESULTS

> **Theorem.** *Assume matrices $m_B \in \mathbb{R}^{p \times r}$ and $m_A \in \mathbb{R}^{r \times q}$ are both full rank. For the objective $\min_{m_B, m_A} L(m_B, m_A; g) = \frac{1}{2} \cdot \|m_B m_A - g\|_F^2$, Newton's method yields the following closed-form update rules:*
>
> $$m_B \leftarrow (1 - \gamma_1) \cdot m_B + \gamma_1 \cdot g m_A^{\ T}(m_A m_A^T)^{-1}, \tag{18}$$
>
> $$m_A \leftarrow (1 - \gamma_1) \cdot m_A + \gamma_1 \cdot (m_B^T m_B)^{-1} m_B^T g. \tag{19}$$
>
> *Here, $\gamma_1$ is the learning rate for the factorized optimization problem.*

*Proof.* We aim to derive Newton's method update rules for the optimization problem $\min_{m_B, m_A} L(m_B, m_A; g) = \frac{1}{2} \cdot \|m_B m_A - g\|_F^2$. Our approach begins with computing the first-order gradients, then proceeds to the Hessian computation, and finally establishes the connection to exponential moving average (EMA) updates. To start, we compute the first-order partial derivatives:

$$\frac{\partial L}{\partial m_B} = (m_B m_A - g) m_A^T, \tag{20}$$

$$\frac{\partial L}{\partial m_A} = m_B^T (m_B m_A - g). \tag{21}$$

While standard gradient descent would directly use these gradients to update the parameters, we instead pursue Newton's method because it yields a more elegant form that naturally resembles EMA updates. For Newton's method, we need the second-order derivatives (Hessian matrices). Computing these second-order partial derivatives gives us:

$$H_{BB} = \frac{\partial^2 L}{\partial m_B^2} = \frac{\partial \left[ (m_B m_A - g) \, m_A^T \right]}{\partial m_B} = m_A m_A^T \otimes I_p, \tag{22}$$

$$H_{AA} = \frac{\partial^2 L}{\partial m_A^2} = \frac{\partial \left[ m_B^T (m_B m_A - g) \right]}{\partial m_A} = I_q \otimes m_B^T m_B. \tag{23}$$

Using these Hessian matrices, we can now compute the Newton directions by solving the linear systems $H \cdot d = \nabla L$, which yields:

$$\text{vec}(d_{m_B}) = H_{BB}^{-1} \text{vec} \left( \frac{\partial L}{\partial m_B} \right) \implies d_{m_B} = m_B - g m_A^T (m_A m_A^T)^{-1}, \tag{24}$$

$$\text{vec}(d_{m_A}) = H_{AA}^{-1} \text{vec} \left( \frac{\partial L}{\partial m_A} \right) \implies d_{m_A} = m_A - (m_B^T m_B)^{-1} m_B^T g. \tag{25}$$

The key insight emerges when we apply these Newton directions with learning rate $\gamma_1$. Substituting the Newton directions into the update formula $x \leftarrow x - \gamma_1 d_x$, we obtain:

$$
\begin{aligned}
m_B &\leftarrow m_B - \gamma_1 \cdot d_{m_B} \\
&= m_B - \gamma_1 \cdot \left[ m_B - g m_A^T (m_A m_A^T)^{-1} \right] \\
&= (1 - \gamma_1) \cdot m_B + \gamma_1 \cdot g m_A^T (m_A m_A^T)^{-1}, \\
m_A &\leftarrow m_A - \gamma_1 \cdot d_{m_A} \\
&= m_A - \gamma_1 \cdot \left[ m_A - (m_B^T m_B)^{-1} m_B^T g \right] \\
&= (1 - \gamma_1) \cdot m_A + \gamma_1 \cdot (m_B^T m_B)^{-1} m_B^T g.
\end{aligned}
$$

(26)

(27)

These final expressions reveal the remarkable property that Newton's method naturally produces update rules in the form of exponential moving averages, where each new parameter value is a weighted combination of the previous value and a target value derived from the optimization objective.

To further illustrate this connection, we note that in the uncompressed case where we optimize $\min_m L(m; g) = \frac{1}{2} \| m - g \|_F^2$, Newton's method similarly yields the classic EMA update:

$$
\begin{aligned}
m &\leftarrow m - \gamma \cdot H_{mm}^{-1} \frac{\partial L}{\partial m} \\
&= (1 - \gamma) \cdot m + \gamma \cdot g.
\end{aligned}
$$

(28)

This consistency across problem formulations demonstrates the fundamental nature of this EMA-like structure in Newton's method and justifies our preference for this approach over standard gradient descent.

$\square$

# B  DETAILED ALGORITHMS OF LORA-PRE FOR ADAM AND MUON OPTIMIZERS

This section presents the LoRA-Pre algorithms for both Adam (Kingma & Ba, 2015) and Muon (Jordan et al., 2024) optimizers.

## B.1  ALGORITHM OF LORA-PRE FOR ADAM

The Adam optimizer update rules under LoRA-Pre have been established in Section 3.

**First-order momentum updates**: For the first-order momentum term with parameterization $m = m_B m_A$, the update rules are:

$$
m_B' \leftarrow (1 - \gamma_1) \cdot m_B + \gamma_1 \cdot g {m_A}^T (m_A m_A^T)^{-1},
$$

(29)

$$
m_A' \leftarrow (1 - \gamma_1) \cdot m_A + \gamma_1 \cdot (m_B^T m_B)^{-1} m_B^T g.
$$

(30)

By default, we set $1 - \gamma_1 = \sqrt{\beta_1}$, which ensures that after the update, $m' = m_B' m_A' = \beta_1 \cdot m_B m_A + \cdots$, making the EMA coefficient consistent with standard Adam.

**Second-order momentum updates**: For the second-order momentum term with parameterization $v = (v_B v_A)^{\circ 2}$, the update rules are:

$$
v_B' \leftarrow (1 - \gamma_2) \cdot v_B + \gamma_2 \cdot |g| {v_A}^T (v_A v_A^T)^{-1},
$$

(31)

$$
v_A' \leftarrow (1 - \gamma_2) \cdot v_A + \gamma_2 \cdot (v_B^T v_B)^{-1} v_B^T |g| .
$$

(32)

Analogously, we set $1 - \gamma_2 = \beta_2^{0.25}$ by default, which ensures that $v' = (v_B' v_A')^{\circ 2} = \beta_2 \cdot (v_B v_A)^{\circ 2} + \cdots$.

**Complete algorithm**: Based on these update formulas, Algorithm 1 presents the complete LoRA-Pre implementation for the Adam optimizer, demonstrating how these factorized momentum updates integrate seamlessly into the standard Adam framework.

---

**Algorithm 1** Comparison of  Adam  and  Adam with LoRA-Pre

---

**Require:** Initial learning rate $\gamma$, weight decay $\lambda$, $\beta_1, \beta_2 \in [0, 1)$, $\gamma_1, \gamma_2 \in [0, 1)$, $\epsilon > 0$

1: Initialize parameters $\theta_0$, time step $t \leftarrow 0$,

  first moment $m_0 \leftarrow 0$, second moment $v_0 \leftarrow 0$ ,

  first low-rank moment $m_{B,0} \leftarrow 0$, $m_{A,0} \sim \mathcal{N}(0, 0.02)$,

  second low-rank moment $v_{B,0} \leftarrow 0$, $v_{A,0} \sim \mathcal{N}(0, 0.02)$.

2: **repeat**

3:    $t \leftarrow t + 1$

4:    $g_t \leftarrow \nabla_\theta \mathcal{L}_t(\theta_{t-1})$

5:    # Update first moment

6:    $m_t \leftarrow \beta_1 \cdot m_{t-1} + (1 - \beta_1) \cdot g_t$

7:    $m_t \leftarrow \beta_1 \cdot m_{B,t-1} m_{A,t-1} + (1 - \beta_1) \cdot g_t$

8:    $m_{B,t} \leftarrow (1 - \gamma_1) \cdot m_{B,t-1} + \gamma_1 \cdot g_t m_{A,t-1}^T (m_{A,t-1} m_{A,t-1}^T)^{-1}$

9:    $m_{A,t} \leftarrow (1 - \gamma_1) \cdot m_{A,t-1} + \gamma_1 \cdot (m_{B,t-1}^T m_{B,t-1})^{-1} m_{B,t-1}^T g_t$

10:   # Update second moment

11:   $v_t \leftarrow \beta_2 \cdot v_{t-1} + (1 - \beta_2) \cdot g_t^{\circ 2}$

12:   $v_t \leftarrow \beta_2 \cdot (v_{B,t-1} v_{A,t-1})^{\circ 2} + (1 - \beta_2) \cdot g_t^{\circ 2}$

13:   $v_{B,t} \leftarrow (1 - \gamma_2) \cdot v_{B,t-1} + \gamma_2 \cdot |g_t| \cdot v_{A,t-1}^T (v_{A,t-1} v_{A,t-1}^T)^{-1}$

14:   $v_{A,t} \leftarrow (1 - \gamma_2) \cdot v_{A,t-1} + \gamma_2 \cdot (v_{B,t-1}^T v_{B,t-1})^{-1} v_{B,t-1}^T \cdot |g_t|$

15:   $\hat{m}_t \leftarrow m_t / (1 - \beta_1^t)$

16:   $\hat{v}_t \leftarrow v_t / (1 - \beta_2^t)$

17:   $\theta_t \leftarrow \theta_{t-1} - \gamma \left( \frac{\hat{m}_t}{\sqrt{\hat{v}_t} + \epsilon} + \lambda \theta_{t-1} \right)$

18: **until** stopping criterion is met

19: **return** Optimized parameters $\theta_t$

---

## B.2 Algorithm of LoRA-Pre for Muon

In this section, we present the LoRA-Pre algorithm for the Muon (Jordan et al., 2024) optimizer.

**First-order momentum updates**: For the Muon optimizer, we derive the LoRA-Pre algorithm by first reformulating the momentum update. The Muon momentum term can be equivalently written as:

$$
\begin{aligned}
m' &= \mu \cdot m + g \\
&= m - (1 - \mu) \cdot m + g \\
&= m - (1 - \mu) \cdot (m - g) + \mu \cdot g \\
&= m - (1 - \mu) \cdot \left[ (m - g) - \frac{\mu}{1 - \mu} \cdot g \right].
\end{aligned}
\tag{33}
$$

By treating the Muon update as the solution to an optimization problem, we can derive the equivalent objective function:

$$
L(m; g) = \frac{1}{2} \cdot \|m - g\|_F^2 - \frac{\mu}{1 - \mu} \cdot \langle m, g \rangle_F.
\tag{34}
$$

After applying low-rank factorization $m = m_B \cdot m_A$, the objective becomes:

$$
L(m_B, m_A; g) = \frac{1}{2} \cdot \|m_B m_A - g\|_F^2 - \frac{\mu}{1 - \mu} \cdot \langle m_B m_A, g \rangle_F.
\tag{35}
$$

We aim to derive Newton's method update rules for this modified objective. Computing the first-order gradients:

$$
\frac{\partial L}{\partial m_B} = (m_B m_A - g) m_A^T - \frac{\mu}{1 - \mu} \cdot g m_A^T,
\tag{36}
$$

$$
\frac{\partial L}{\partial m_A} = m_B^T (m_B m_A - g) - \frac{\mu}{1 - \mu} \cdot m_B^T g.
\tag{37}
$$

The Hessian matrices have the same structure as before since the additional linear term doesn't affect the second derivatives:

$$
H_{BB} = \frac{\partial^2 L}{\partial m_B^2} = \frac{\partial \left[ (m_B m_A - g) m_A^T \right]}{\partial m_B} = m_A m_A^T \otimes I_p,
\tag{38}
$$

$$
H_{AA} = \frac{\partial^2 L}{\partial m_A^2} = \frac{\partial \left[ m_B^T (m_B m_A - g) \right]}{\partial m_A} = I_q \otimes m_B^T m_B.
\tag{39}
$$

Using these Hessian matrices, we can now compute the Newton directions by solving the linear systems $H \cdot d = \nabla L$, which yields:

$$
\text{vec}(d_{m_B}) = H_{BB}^{-1} \text{vec}\left( \frac{\partial L}{\partial m_B} \right) \implies d_{m_B} = m_B - \frac{1}{1 - \mu} \cdot g m_A^T (m_A m_A^T)^{-1},
\tag{40}
$$

$$
\text{vec}(d_{m_A}) = H_{AA}^{-1} \text{vec}\left( \frac{\partial L}{\partial m_A} \right) \implies d_{m_A} = m_A - \frac{1}{1 - \mu} \cdot (m_B^T m_B)^{-1} m_B^T g.
\tag{41}
$$

The key insight emerges when we apply these Newton directions with learning rate $\gamma_1$. Substituting the Newton directions into the update formula $x \leftarrow x - \gamma_1 \cdot d_x$, we obtain:

$$
\begin{aligned}
m_B &\leftarrow m_B - \gamma_1 \cdot d_{m_B} \\
&= m_B - \gamma_1 \cdot \left[ m_B - \frac{1}{1 - \mu} \cdot g m_A^T (m_A m_A^T)^{-1} \right] \\
&= (1 - \gamma_1) \cdot m_B + \frac{\gamma_1}{1 - \mu} \cdot g m_A^T (m_A m_A^T)^{-1},
\end{aligned}
\tag{42}
$$

$$
\begin{aligned}
m_A &\leftarrow m_A - \gamma_1 \cdot d_{m_A} \\
&= m_A - \gamma_1 \cdot \left[ m_A - \frac{1}{1 - \mu} \cdot (m_B^T m_B)^{-1} m_B^T g \right] \\
&= (1 - \gamma_1) \cdot m_A + \frac{\gamma_1}{1 - \mu} \cdot (m_B^T m_B)^{-1} m_B^T g.
\end{aligned}
\tag{43}
$$

Similarly, we set $1 - \gamma_1 = \sqrt{\mu}$.

**Complete algorithm**: Based on these update formulas, Algorithm 2 presents the complete LoRA-Pre implementation for the Muon optimizer, demonstrating how these factorized momentum updates integrate seamlessly into the standard Muon framework.

---

**Algorithm 2** Comparison of Muon and Muon with LoRA-Pre

---

**Require:** Initial learning rate $\gamma$, weight decay $\lambda$, momentum $\mu \in [0,1)$, $\gamma_1 \in [0,1)$

1: Initialize parameters $\theta_0$, time step $t \leftarrow 0$,

   first moment $m_0 \leftarrow 0$,

   first low-rank moment $m_{B,0} \leftarrow 0$, $m_{A,0} \sim \mathcal{N}(0, 0.02)$.

2: **repeat**

3:      $t \leftarrow t + 1$

4:      $g_t \leftarrow \nabla_\theta \mathcal{L}_t(\theta_{t-1})$

5:      # Update first moment

6:      $m_t \leftarrow \mu \cdot m_{t-1} + g_t$

7:      $m_t \leftarrow \mu \cdot m_{B,t-1} m_{A,t-1} + g_t$

8:      $m_{B,t} \leftarrow (1 - \gamma_1) \cdot m_{B,t-1} + \frac{\gamma_1}{1-\mu} \cdot g_t m_{A,t-1}^T (m_{A,t-1} m_{A,t-1}^T)^{-1}$

9:      $m_{A,t} \leftarrow (1 - \gamma_1) \cdot m_{A,t-1} + \frac{\gamma_1}{1-\mu} \cdot (m_{B,t-1}^T m_{B,t-1})^{-1} m_{B,t-1}^T g_t$

10:     $O_t \leftarrow \text{NewtonSchulz5}(m_t)$

11:     $\theta_t \leftarrow \theta_{t-1} - \gamma \left( O_t + \lambda \theta_{t-1} \right)$

12: **until** stopping criterion is met

13: **return** Optimized parameters $\theta_t$

---

# C THEORETICAL ANALYSIS OF APPROXIMATION ERROR AND CONVERGENCE

In this appendix, we provide a rigorous theoretical analysis of the **LoRA-Pre Adam** optimizer. We explicitly analyze the approximation error introduced by the low-rank factorization of the optimizer states, and prove the convergence fidelity of the algorithm in non-convex settings.

## C.1 PROBLEM SETUP AND ALGORITHM DYNAMICS

Consider the unconstrained optimization problem $\min_{\theta \in \mathbb{R}^{p \times q}} f(\theta)$. Let $g_t = \nabla f(\theta_{t-1})$ be the stochastic gradient at step $t$. We denote the states of **Standard Adam** as $m_t$, $v_t$ and the effective states of **LoRA-Pre Adam** as $\tilde{m}_t$, $\tilde{v}_t$.

**1. Standard Adam Dynamics**   The standard optimizer updates its moments using exponential moving averages (EMA) with decay rates $\beta_1, \beta_2 \in [0,1)$:

$$m_t = \beta_1 \cdot m_{t-1} + (1 - \beta_1) \cdot g_t, \tag{44}$$

$$v_t = \beta_2 \cdot v_{t-1} + (1 - \beta_2) \cdot g_t^{\circ 2}. \tag{45}$$

**2. LoRA-Pre Dynamics**   LoRA-Pre maintains low-rank factors $(m_{B,t}, m_{A,t})$ to approximate the gradient history. Let $\gamma_1$ be the update rate. The exact simultaneous update rules (Online Least Squares) can be compactly expressed using the Moore-Penrose pseudo-inverse $(\cdot)^\dagger$:

$$m_{B,t} = (1 - \gamma_1) \cdot m_{B,t-1} + \gamma_1 \cdot g_t m_{A,t-1}^\dagger, \tag{46}$$

$$m_{A,t} = (1 - \gamma_1) \cdot m_{A,t-1} + \gamma_1 \cdot m_{B,t-1}^\dagger g_t, \tag{47}$$

where the Tikhonov regularized pseudo-inverses for the full-rank factors are defined as $m_A^\dagger = m_A^T (m_A m_A^T + \lambda \cdot I)^{-1}$ and $m_B^\dagger = (m_B^T m_B + \lambda \cdot I)^{-1} m_B^T$ with damping factor $\lambda > 0$.

We define the canonical projection operators associated with the factors at step $t - 1$:

- $\mathcal{P}_A \triangleq m_{A,t-1}^\dagger m_{A,t-1}$ (Projection onto Row Space of $m_A$).

- $\mathcal{P}_B \triangleq m_{B,t-1} m_{B,t-1}^\dagger$ (Projection onto Column Space of $m_B$).

Let $\widehat{m}_t = m_{B,t} m_{A,t}$ be the low-rank history reconstruction. Analogous updates apply to the second moment factors using the gradient magnitude $|g_t|$, producing the reconstruction $\widehat{h}_t = v_{B,t} v_{A,t}$.

**3. Effective Moments for Update**   Crucially, LoRA-Pre computes the effective moments for the parameter update by combining the low-rank history with the *exact* current gradient:

$$\tilde{m}_t = \beta_1 \cdot \widehat{m}_{t-1} + (1 - \beta_1) \cdot g_t, \tag{48}$$

$$\tilde{v}_t = \beta_2 \cdot (\widehat{h}_{t-1})^{\circ 2} + (1 - \beta_2) \cdot g_t^{\circ 2}. \tag{49}$$

Note that for the second moment, LoRA-Pre approximates the history of magnitudes $\widehat{h}$ and then squares it.

## C.2   Assumptions

**Assumption 1** (Regularity and Boundedness). *The objective function and stochastic gradients satisfy the following conditions:*

1. ***L-Smoothness:*** *The objective function $f$ is L-smooth: $\|\nabla f(x) - \nabla f(y)\|_F \leq L \|x - y\|_F$.*

2. ***Bounded Gradients:*** *The stochastic gradients are uniformly bounded in both Frobenius and infinity norms. There exist constants $G$ and $G_\infty$ such that for all $t$, $\|g_t\|_F \leq G$ and $\|g_t\|_\infty \leq G_\infty$.*

3. ***Lipschitz Smooth Update:*** *The optimizer uses a smoothed damping term $\epsilon > 0$. The update mapping is defined as $\phi(m, v) = \frac{m}{\sqrt{v} + \epsilon}$. This function is Lipschitz continuous with respect to both arguments, with constants $L_m = \frac{1}{\sqrt{\epsilon}}$ and $L_v = \frac{G}{2\epsilon^{1.5}}$.*

**Assumption 2** (Subspace Approximation Capability). *The gradient dynamics admit a low-rank structure. Crucially, we assume this structure holds for both the gradient direction and its elementwise magnitude. Let $\mathcal{P}_{B,t}, \mathcal{P}_{A,t}$ denote the projections onto the subspaces maintained by the optimizer at step $t$. We assume there exists a bound $\delta \geq 0$ such that:*

$$\|g_t - (\mathcal{P}_{B,t} g_t + g_t \mathcal{P}_{A,t})\|_F \leq \delta, \tag{50}$$

$$\||g_t| - (\mathcal{P}_{B,t} |g_t| + |g_t| \mathcal{P}_{A,t})\|_F \leq \delta. \tag{51}$$

*The second inequality ensures that the second-moment estimator (based on $|g_t|$) also admits a bounded reconstruction error.*

**Assumption 3** (Reference Optimizer Descent). *Let $u_t = m_t / (\sqrt{v_t} + \epsilon)$ be the update direction of the standard full-rank Adam optimizer. We assume that in expectation, $u_t$ is a valid descent direction aligned with the true gradient:*

$$\mathbb{E}[\langle \nabla f(\theta_t), u_t \rangle] \geq c \mathbb{E}[\|\nabla f(\theta_t)\|_F^2], \tag{52}$$

*for some constant $c > 0$. This assumption anchors the convergence of LoRA-Pre Adam to the theoretical behavior of standard Adam.*

## C.3   Boundedness of Factor Reconstruction Error

We first prove that the error of the stored low-rank history $\widehat{m}_t$ is uniformly bounded. We strictly enforce the time-scale alignment condition: $\beta_1 = (1 - \gamma_1)^2$.

**Lemma C.1.** *Let $\mathcal{E}_t^m = \|m_t - \widehat{m}_t\|_F$. Under Assumptions 1 and 2, $\mathcal{E}_t^m$ is uniformly bounded by a constant $\mathcal{E}_{bound}$.*

*Proof.* **Step 1: Exact Expansion of LoRA Dynamics** Substitute the update rules (46) and (47) into $\widehat{m}_t = m_{B,t}m_{A,t}$:

$$\widehat{m}_t = \left[(1-\gamma_1)m_{B,t-1} + \gamma_1 g_t m_{A,t-1}^\dagger\right] \cdot \left[(1-\gamma_1)m_{A,t-1} + \gamma_1 m_{B,t-1}^\dagger g_t\right]$$

$$= (1-\gamma_1)^2 \widehat{m}_{t-1} + \gamma_1(1-\gamma_1)(\mathcal{P}_B g_t + g_t \mathcal{P}_A) + \gamma_1^2 Q_t, \tag{53}$$

where $Q_t = g_t m_{A,t-1}^\dagger m_{B,t-1}^\dagger g_t$ is the quadratic interaction term. Using the condition $\beta_1 = (1-\gamma_1)^2$, this implies $\gamma_1 = 1 - \sqrt{\beta_1}$. The expansion becomes:

$$\widehat{m}_t = \beta_1 \widehat{m}_{t-1} + (1-\sqrt{\beta_1})\sqrt{\beta_1}(\mathcal{P}_B g_t + g_t \mathcal{P}_A) + (1-\sqrt{\beta_1})^2 Q_t. \tag{54}$$

**Step 2: Constructing the Recursive Error** We form the difference with the standard Adam update $m_t = \beta_1 m_{t-1} + (1-\beta_1)g_t$:

$$m_t - \widehat{m}_t = \beta_1(m_{t-1} - \widehat{m}_{t-1}) + R_t, \tag{55}$$

where the residual driving term $R_t$ is:

$$R_t = (1-\beta_1)g_t - (1-\sqrt{\beta_1})\sqrt{\beta_1}(\mathcal{P}_B g_t + g_t \mathcal{P}_A) - (1-\sqrt{\beta_1})^2 Q_t. \tag{56}$$

**Step 3: Bounding the Residual Magnitude**

We explicitly bound the norm of the driving residual $R_t$. Applying the triangle inequality to the residual definition (56), we decompose the total bound into linear and quadratic contributions:

$$\|R_t\|_F \leq \underbrace{\left\|(1-\sqrt{\beta_1})\left[g_t + \sqrt{\beta_1}(g_t - \mathcal{P}_B g_t - g_t \mathcal{P}_A)\right]\right\|_F}_{\text{Linear Term}} + \underbrace{(1-\sqrt{\beta_1})^2 \|Q_t\|_F}_{\text{Quadratic Term}}. \tag{57}$$

*1. The Linear Term:* Using the homogeneity of the norm and the triangle inequality, followed by the substitution of the gradient bound $\|g_t\|_F \leq G$ (Assumption 1) and the subspace residual bound $\delta$ (Assumption 2), we derive:

$$\|\text{Linear}\|_F = (1-\sqrt{\beta_1})\left\|g_t + \sqrt{\beta_1}(g_t - \mathcal{P}_B g_t - g_t \mathcal{P}_A)\right\|_F$$

$$\leq (1-\sqrt{\beta_1})\left(\|g_t\|_F + \sqrt{\beta_1}\|g_t - \mathcal{P}_B g_t - g_t \mathcal{P}_A\|_F\right)$$

$$\leq (1-\sqrt{\beta_1})G + \sqrt{\beta_1}(1-\sqrt{\beta_1})\delta. \tag{58}$$

*2. The Quadratic Term:* Bounding $\|Q_t\|_F$ requires the spectral norm of the regularized pseudo-inverses. As established in the setup, $m_A^\dagger = m_A^T(m_A m_A^T + \lambda I)^{-1}$ and $m_B^\dagger = (m_B^T m_B + \lambda I)^{-1}m_B^T$. Let the Singular Value Decomposition (SVD) of the factor be $U\Sigma V^T$. Substituting this into either inverse form reveals that the singular values are governed by $h(\sigma) = \frac{\sigma}{\sigma^2 + \lambda}$. Analyzing the function $f(x) = \frac{x}{x^2 + \lambda}$ for $x \geq 0$, we find its maximum at $x = \sqrt{\lambda}$, yielding the uniform spectral bound $\|M^\dagger\|_2 \leq \frac{1}{2\sqrt{\lambda}}$.

Using the sub-multiplicativity of the Frobenius norm (specifically $\|ABC\|_F \leq \|A\|_F \|B\|_2 \|C\|_F$), we strictly bound $Q_t$:

$$\|Q_t\|_F = \|g_t m_{A,t-1}^\dagger m_{B,t-1}^\dagger g_t\|_F$$

$$\leq \|g_t\|_F \cdot \|m_{A,t-1}^\dagger\|_2 \cdot \|m_{B,t-1}^\dagger\|_2 \cdot \|g_t\|_F$$

$$= \|g_t\|_F^2 \cdot \|m_{A,t-1}^\dagger\|_2 \cdot \|m_{B,t-1}^\dagger\|_2$$

$$\leq G^2 \left(\frac{1}{2\sqrt{\lambda}}\right)^2 = \frac{G^2}{4\lambda} \triangleq C_Q. \tag{59}$$

Substituting both bounds back into the residual equation:

$$\|R_t\|_F \leq (1-\sqrt{\beta_1})G + \sqrt{\beta_1}(1-\sqrt{\beta_1})\delta + (1-\sqrt{\beta_1})^2 C_Q \triangleq \Delta_{res}. \tag{60}$$

**Step 4: Convergence to Steady State**

The error dynamics follow the linear recursion inequality $\mathcal{E}_t^m \leq \beta_1 \mathcal{E}_{t-1}^m + \Delta_{res}$. Since the momentum parameter satisfies $0 \leq \beta_1 < 1$, this geometric series converges to a finite steady state as $t \to \infty$:

$$\lim_{t \to \infty} \mathcal{E}_t^m \leq \frac{\Delta_{res}}{1 - \beta_1} \triangleq \mathcal{E}_{bound}. \tag{61}$$

To derive the explicit bound, we substitute the expression for $\Delta_{res}$ derived in Equation (60) and factor the denominator using $1 - \beta_1 = (1 - \sqrt{\beta_1})(1 + \sqrt{\beta_1})$:

$$
\begin{aligned}
\mathcal{E}_{bound} &= \frac{(1 - \sqrt{\beta_1})G + \sqrt{\beta_1}(1 - \sqrt{\beta_1})\delta + (1 - \sqrt{\beta_1})^2 C_Q}{(1 - \sqrt{\beta_1})(1 + \sqrt{\beta_1})} \\
&= \frac{(1 - \sqrt{\beta_1})\left[G + \sqrt{\beta_1}\delta + (1 - \sqrt{\beta_1})C_Q\right]}{(1 - \sqrt{\beta_1})(1 + \sqrt{\beta_1})}.
\end{aligned} \tag{62}
$$

Canceling the common factor $(1 - \sqrt{\beta_1})$ from the numerator and denominator, we obtain the final uniform bound:

$$\mathcal{E}_{bound} = \frac{G + \sqrt{\beta_1}\delta + (1 - \sqrt{\beta_1})C_Q}{1 + \sqrt{\beta_1}}. \tag{63}$$

This confirms that the reconstruction error is uniformly bounded by a constant determined by the gradient magnitude $G$, the subspace residual $\delta$, and the quadratic interaction $C_Q$. $\qquad\square$

### C.4 JOINT EFFECTIVE MOMENT ERROR

We now derive the error bounds for the effective moments $\tilde{m}_t$ and $\tilde{v}_t$ used in the parameter update, explicitly accounting for the non-linear square term in $\tilde{v}_t$.

**Lemma C.2.** *Let $\Delta_m = \|m_t - \tilde{m}_t\|_F$ and $\Delta_v = \|v_t - \tilde{v}_t\|_F$ denote the effective moment errors. Let $d$ be the total number of parameters. The errors are bounded by:*

$$\Delta_m \leq \beta_1 \mathcal{E}_{bound}, \tag{64}$$

$$\Delta_v \leq \beta_2 \left(2G_\infty \mathcal{E}_{bound} + \sigma_{total}^2\right), \tag{65}$$

*where $\sigma_{total}^2 = \frac{\sqrt{d}}{4}G_\infty^2$ represents the intrinsic variance bound derived from Popoviciu's inequality.*

*Proof.* **1. First Moment Error Analysis**

We compare the effective first moment $\tilde{m}_t$ with the standard Adam moment $m_t$. Subtracting their definitions:

$$m_t - \tilde{m}_t = \beta_1(m_{t-1} - \widehat{m}_{t-1}) + (1 - \beta_1)(g_t - g_t) = \beta_1(m_{t-1} - \widehat{m}_{t-1}). \tag{66}$$

The current gradient terms cancel out. Taking the Frobenius norm and applying Lemma C.1, we obtain:

$$\Delta_m \leq \beta_1 \mathcal{E}_{bound}. \tag{67}$$

**2. Second Moment Error Analysis**

We analyze the error in the second moment estimate. Standard Adam updates the second moment as $v_t = \beta_2 v_{t-1} + (1 - \beta_2)g_t^{\circ 2}$. LoRA-Pre updates the effective second moment as $\tilde{v}_t = \beta_2(\widehat{h}_{t-1})^{\circ 2} + (1 - \beta_2)g_t^{\circ 2}$.

Subtracting the two equations, the term $(1 - \beta_2)g_t^{\circ 2}$ involving the current gradient cancels out exactly. The error is strictly propagated from the history terms:

$$v_t - \tilde{v}_t = \beta_2 \left[v_{t-1} - (\widehat{h}_{t-1})^{\circ 2}\right]. \tag{68}$$

To rigorously bound this, we introduce an auxiliary variable $h_{t-1}$, defined as the **exact** exponential moving average of the gradient magnitudes $|g|$ (i.e., $h_k = \beta_2 h_{k-1} + (1 - \beta_2)|g_k|$). We apply the triangle inequality to decompose the error into two structurally distinct terms:

$$\Delta_v \leq \beta_2 \left( \underbrace{\|v_{t-1} - (h_{t-1})^{\circ 2}\|_F}_{\text{Term I: Intrinsic Variance}} + \underbrace{\|(h_{t-1})^{\circ 2} - (\widehat{h}_{t-1})^{\circ 2}\|_F}_{\text{Term II: Approximation Error}} \right). \tag{69}$$

*Term I: Intrinsic Variance (Popoviciu's Bound).* This term captures the discrepancy between "the mean of squares" and "the square of the mean". Let us focus on a single parameter index $j$. The EMA formulation can be interpreted as a weighted expectation $\mathbb{E}_w$ over the history of gradients, where the weights sum to 1. Thus:

$$v_{t-1,j} = \mathbb{E}_w[|g_{\tau,j}|^2] \quad \text{and} \quad (h_{t-1,j})^2 = (\mathbb{E}_w[|g_{\tau,j}|])^2. \tag{70}$$

The difference $v_{t-1,j} - (h_{t-1,j})^2$ is precisely the variance of the random variable $|g_{\tau,j}|$ under the EMA probability measure. Since the gradient magnitude is bounded, i.e., $|g_{\tau,j}| \in [0, G_\infty]$, we invoke Popoviciu's Inequality, which states that the variance of any bounded random variable on interval $[a, b]$ is bounded by $\frac{1}{4}(b - a)^2$. Here, $[a, b] = [0, G_\infty]$. Thus, for every element $j$:

$$\left| v_{t-1,j} - (h_{t-1,j})^2 \right| \leq \frac{1}{4}(G_\infty - 0)^2 = \frac{1}{4}G_\infty^2. \tag{71}$$

We generalize this to the full parameter matrix by computing the Frobenius norm over all $d$ elements:

$$\|v_{t-1} - (h_{t-1})^{\circ 2}\|_F = \sqrt{\sum_{j=1}^{d} |v_{t-1,j} - (h_{t-1,j})^2|^2} \leq \sqrt{d \cdot \left(\frac{1}{4}G_\infty^2\right)^2} = \frac{\sqrt{d}}{4}G_\infty^2. \tag{72}$$

We define this constant bound as $\sigma_{total}^2 \triangleq \frac{\sqrt{d}}{4}G_\infty^2$.

*Term II: Approximation Error.* This term represents the propagation of the low-rank reconstruction error through the squaring operation. Consider the function $f(x) = x^2$. For inputs restricted to the domain $[0, G_\infty]$, the derivative is bounded by $|f'(x)| = |2x| \leq 2G_\infty$. By the Mean Value Theorem, the function is Lipschitz continuous with constant $L_{sq} = 2G_\infty$. Applying this element-wise to the vectors $h_{t-1}$ and $\widehat{h}_{t-1}$:

$$\left| (h_{t-1,j})^2 - (\widehat{h}_{t-1,j})^2 \right| \leq 2G_\infty \left| h_{t-1,j} - \widehat{h}_{t-1,j} \right|. \tag{73}$$

Squaring both sides and summing over all $j$ to recover the Frobenius norm:

$$\sum_{j=1}^{d} \left| (h_{t-1,j})^2 - (\widehat{h}_{t-1,j})^2 \right|^2 \leq 4G_\infty^2 \sum_{j=1}^{d} \left| h_{t-1,j} - \widehat{h}_{t-1,j} \right|^2. \tag{74}$$

Taking the square root yields:

$$\|(h_{t-1})^{\circ 2} - (\widehat{h}_{t-1})^{\circ 2}\|_F \leq 2G_\infty \|h_{t-1} - \widehat{h}_{t-1}\|_F. \tag{75}$$

Substituting the bound from Lemma C.1 ($\|h_{t-1} - \widehat{h}_{t-1}\|_F \leq \mathcal{E}_{bound}$), we have:

$$\text{Term II} \leq 2G_\infty \mathcal{E}_{bound}. \tag{76}$$

Combining the bounds for Term I and Term II, we obtain the final bound for the second moment error:

$$\Delta_v \leq \beta_2 \left(\sigma_{total}^2 + 2G_\infty \mathcal{E}_{bound}\right). \tag{77}$$

$\square$

## C.5 CONVERGENCE ANALYSIS

**Theorem C.3.** *Let the step size be $\eta_t = \eta/\sqrt{t}$. Under Assumptions 1, 2, and 3, LoRA-Pre Adam (with update rule $\phi(m, v) = m/\sqrt{v + \epsilon}$) converges to a neighborhood of a stationary point:*

$$\min_{1 \leq t \leq T} \mathbb{E}[\|\nabla f(\theta_t)\|^2] \leq \frac{C_{init}}{\sqrt{T}} + C_{noise} \left(\mathcal{E}_{bound} + \sigma_{total}^2\right)^2, \tag{78}$$

*where $C_{init}$ depends on the initial function gap and $C_{noise}$ depends on the Lipschitz constants of the update rule.*

*Proof.* **1. Bounding the Update Direction Error**

Let $u_t = \phi(m_t, v_t)$ and $\tilde{u}_t = \phi(\tilde{m}_t, \tilde{v}_t)$ be the ideal and actual update directions, respectively. We analyze the theoretical update function $\phi(m, v) = \frac{m}{\sqrt{v+\epsilon}}$ (using the smoothed denominator for Lipschitz continuity). The partial derivatives are bounded as follows:

- W.r.t $m$: $\|\nabla_m \phi\| \leq \frac{1}{\sqrt{\epsilon}} \triangleq L_m$.

- W.r.t $v$: $\|\nabla_v \phi\| \leq \sup_{v \geq 0} \left\| \frac{-m}{2(v+\epsilon)^{1.5}} \right\| \leq \frac{G}{2\epsilon^{1.5}} \triangleq L_v$.

Using the moment error bounds from Lemma C.2, the total direction error $\xi_t = \|u_t - \tilde{u}_t\|_F$ satisfies:

$$
\begin{aligned}
\xi_t &\leq L_m \Delta_m + L_v \Delta_v \\
&\leq \frac{1}{\sqrt{\epsilon}} (\beta_1 \mathcal{E}_{bound}) + \frac{G}{2\epsilon^{1.5}} \beta_2 (2G_\infty \mathcal{E}_{bound} + \sigma_{total}^2) \\
&\leq \underbrace{\left( \frac{\beta_1}{\sqrt{\epsilon}} + \frac{GG_\infty \beta_2}{\epsilon^{1.5}} \right)}_{K_1} \mathcal{E}_{bound} + \underbrace{\left( \frac{G\beta_2}{2\epsilon^{1.5}} \right)}_{K_2} \sigma_{total}^2 \triangleq \xi_{total}.
\end{aligned}
\tag{79}
$$

Here, $\xi_{total}$ is a uniform deterministic upper bound on the direction error.

**2. One-Step Descent Analysis in Expectation**

Since the objective function $f$ is $L$-smooth, we apply the Descent Lemma conditioned on the current state $\theta_t$:

$$
\mathbb{E}_t[f(\theta_{t+1})] \leq f(\theta_t) - \eta_t \mathbb{E}_t[\langle \nabla f(\theta_t), \tilde{u}_t \rangle] + \frac{L\eta_t^2}{2} \mathbb{E}_t[\|\tilde{u}_t\|^2].
\tag{80}
$$

Let $G_{step} = G/\sqrt{\epsilon}$ be the bound on the update norm $\|\tilde{u}_t\|$. We decompose the inner product term:

$$
-\mathbb{E}_t[\langle \nabla f, \tilde{u}_t \rangle] = -\mathbb{E}_t[\langle \nabla f, u_t \rangle] + \mathbb{E}_t[\langle \nabla f, u_t - \tilde{u}_t \rangle].
\tag{81}
$$

Using Assumption 3 ($\mathbb{E}_t[\langle \nabla f, u_t \rangle] \geq c\|\nabla f\|^2$) and applying Cauchy-Schwarz followed by Young's Inequality ($ab \leq \frac{c}{2}a^2 + \frac{1}{2c}b^2$) to the error term:

$$
\begin{aligned}
\mathbb{E}_t[\langle \nabla f, u_t - \tilde{u}_t \rangle] &\leq \|\nabla f\| \xi_{total} \\
&\leq \frac{c}{2}\|\nabla f\|^2 + \frac{1}{2c} \xi_{total}^2.
\end{aligned}
\tag{82}
$$

Substituting this back into the descent inequality, the descent term $-c\|\nabla f\|^2$ is partially offset by the error:

$$
\begin{aligned}
\mathbb{E}_t[f(\theta_{t+1})] &\leq f(\theta_t) - \eta_t \left( c\|\nabla f\|^2 - \frac{c}{2}\|\nabla f\|^2 - \frac{1}{2c}\xi_{total}^2 \right) + \frac{L\eta_t^2}{2} G_{step}^2 \\
&= f(\theta_t) - \frac{c\eta_t}{2}\|\nabla f\|^2 + \frac{\eta_t}{2c}\xi_{total}^2 + \frac{L\eta_t^2}{2} G_{step}^2.
\end{aligned}
\tag{83}
$$

**3. Global Convergence**

Taking the total expectation and summing from $t = 1$ to $T$:

$$
\sum_{t=1}^{T} \frac{c\eta_t}{2} \mathbb{E}[\|\nabla f(\theta_t)\|^2] \leq f(\theta_1) - f^* + \frac{\xi_{total}^2}{2c} \sum_{t=1}^{T} \eta_t + \frac{LG_{step}^2}{2} \sum_{t=1}^{T} \eta_t^2.
\tag{84}
$$

Dividing by $\frac{c}{2} S_T$ (where $S_T = \sum \eta_t$), we isolate the minimum gradient norm:

$$
\min_t \mathbb{E}[\|\nabla f\|^2] \leq \frac{2(f(\theta_1) - f^*)}{cS_T} + \frac{1}{c^2} \xi_{total}^2 + \frac{LG_{step}^2}{c} \frac{\sum \eta_t^2}{S_T}.
\tag{85}
$$

Given $\eta_t = \eta/\sqrt{t}$, we have $S_T \geq \sqrt{T}$ and $\sum \eta_t^2 \approx \ln T$. Thus, as $T \to \infty$, the first and third terms vanish ($O(1/\sqrt{T})$ and $O(\ln T/\sqrt{T})$), leaving the constant error floor. Defining $C_{noise} = \frac{1}{c^2} \max(K_1, K_2)^2$ (noting that $\xi_{total}^2 \leq 2\max(K_1, K_2)^2 (\mathcal{E}_{bound} + \sigma_{total}^2)^2$), we conclude:

$$
\min_t \mathbb{E}[\|\nabla f\|^2] \leq \frac{C_{init}}{\sqrt{T}} + C_{noise}(\mathcal{E}_{bound} + \sigma_{total}^2)^2.
\tag{86}
$$

This confirms that the algorithm converges to a neighborhood determined by the approximation quality $\mathcal{E}_{bound}$ and intrinsic variance $\sigma_{total}^2$.

$\square$

# D  ADDITIONAL EXPERIMENTAL RESULTS

## D.1  HYPERPARAMETER SENSITIVITY ANALYSIS: THE COUPLING OF $\beta$ AND $\gamma$

In this section, we evaluate the sensitivity of LoRA-Pre Adam to variations in its momentum hyper-parameters. A key design principle of LoRA-Pre is that the low-rank update coefficients $(\gamma_1, \gamma_2)$ are *not* independent hyperparameters requiring separate tuning. Instead, they are analytically coupled with the standard Adam momentum coefficients $(\beta_1, \beta_2)$ to ensure consistent decay dynamics.

**Derivation of the Coupling Strategy.** Formally, the update rules for the low-rank momentum components $m_B$ and $m_A$ are defined as:

$$m_B' \leftarrow (1 - \gamma_1) \cdot m_B + \gamma_1 \cdot g m_A^T (m_A m_A^T)^{-1}, \tag{87}$$

$$m_A' \leftarrow (1 - \gamma_1) \cdot m_A + \gamma_1 \cdot (m_B^T m_B)^{-1} m_B^T g. \tag{88}$$

When reconstructing the effective full-rank momentum $m \approx m_B m_A$, the decay behavior can be approximated by expanding the product of the updated factors. Ignoring higher-order cross terms, the effective decay rate is the product of the individual decay rates:

$$m' = m_B' m_A' \approx (1 - \gamma_1)^2 \cdot m_B m_A + \mathcal{O}(\gamma_1^2) \approx (1 - \gamma_1)^2 \cdot m. \tag{89}$$

To align this trajectory with standard Adam optimization (where momentum decays by $\beta_1$), we enforce the coupling constraint $(1 - \gamma_1)^2 = \beta_1$. Similarly, for the second moment, since LoRA-Pre approximates the gradient magnitude $\widehat{h} \approx |g|$ and subsequently squares it ($\tilde{v} = \widehat{h}^{\circ 2}$), the decay rate compounds twice, leading to the condition $(1 - \gamma_2)^4 = \beta_2$. Consequently, the learning dynamics are strictly governed by $\beta$, eliminating the need to grid-search $\gamma$.

**Empirical Verification.** We conducted sensitivity studies on the 60M parameter model by varying $\beta_1$ and $\beta_2$ around their standard default values ($\beta_1 = 0.9, \beta_2 = 0.95$). The results are summarized in Table 4.

Table 4: **Sensitivity Analysis of Momentum Parameters $\beta$.** We report the validation perplexity on the 60M model. The method exhibits stability around the default settings derived from standard Adam ($\beta_1 = 0.9, \beta_2 = 0.95$). Extreme values (e.g., $\beta \to 1$) cause the implicit $\gamma$ to vanish, leading to training instability.

| Parameter | Value | Perplexity | Status |
|---|---|---|---|
| | 0.90 (*Default*) | **32.57** | **Optimal** |
| $\beta_1$ (fix $\beta_2 = 0.95$) | 0.95 | 37.62 | Stable |
| | 0.99 | 1458.92 | Diverged |
| | 0.90 | 34.61 | Stable |
| $\beta_2$ (fix $\beta_1 = 0.90$) | 0.95 (*Default*) | **32.57** | **Optimal** |
| | 0.999 | 1301.58 | Diverged |

As shown in Table 4, LoRA-Pre achieves the best performance at the configuration inherited from standard Adam. While the optimizer is robust within a reasonable range, extreme values (e.g., $\beta_1 = 0.99$ or $\beta_2 = 0.999$) lead to numerical instability. This is expected: as $\beta \to 1$, the derived update rate $\gamma$ approaches zero (e.g., for $\beta_2 = 0.999$, $\gamma_2 \approx 2.5 \times 10^{-4}$), causing the low-rank factors to adapt too slowly to track the shifting gradient subspace. This confirms that our coupling strategy is effective, but the "sweet spot" for low-rank adaptation dynamics aligns with, but is more sensitive than, full-rank optimization.

# E STATEMENT OF THE USE OF LARGE LANGUAGE MODELS

The use of LLMs in this work was restricted to paper writing assistance. They were not used to generate results, derive proofs, or conduct analysis without human verification. The disclosure here, as well as in the submission form, fulfills the ICLR requirement that all contributions of LLMs be acknowledged transparently.

