# OpenReview forum: "Taming Momentum: Rethinking Optimizer States Through Low-Rank Approximation"
_ICLR.cc/2026/Conference — ICLR 2026 Oral_

### Official Review · Reviewer_Mkay · 2025-10-20

**Soundness:** 3
**Presentation:** 3
**Contribution:** 3
**Rating:** 6
**Confidence:** 3

**Summary:**

The paper proposes LoRA-Pre, a new optimizer framework that reinterprets momentum as an online linear regression process and replaces traditional exponential-moving-average (EMA) updates with a low-rank closed-form solution. By decomposing momentum into low-rank factors and updating them analytically, LoRA-Pre significantly reduces optimizer-state memory while preserving training stability. The method is applied to both Adam and Muon, yielding LoRA-Pre-Adam and LoRA-Pre-Muon variants. Experiments on large-scale pretraining (LLaMA-1B and smaller models) and instruction fine-tuning (LLaMA-8B, LLaMA-7B) show that LoRA-Pre matches or surpasses strong baselines such as Galore, DoRA, and ReLoRA, while offering notable memory savings and rank efficiency.

**Strengths:**

The paper introduces a clear and technically sound rethinking of momentum optimization by framing EMA updates as an online regression problem and deriving a low-rank closed-form solution. This perspective is both conceptually original and practically valuable, bridging optimization theory with efficient model training. The proposed LoRA-Pre method is well-integrated with existing optimizers like Adam and Muon, achieving strong empirical results with lower memory costs. The experiments are comprehensive, covering both large-scale pretraining and fine-tuning, and the presentation is mathematically rigorous and well-organized.

**Weaknesses:**

The empirical evaluation primarily compares with projection-based low-rank optimizers; incorporating a wider range of modern baselines (e.g., Sophia, Shampoo) would provide a more complete assessment of practical advantages.

The ablation studies emphasize rank efficiency but do not clearly disentangle the effect of the low-rank momentum representation from other implementation factors such as learning-rate scaling or normalization.

The scalability and computational trade-offs of the proposed updates, particularly under distributed or large-batch training, are not fully analyzed, leaving some uncertainty about deployment efficiency in large-scale settings.

**Questions:**

Can the authors clarify how the proposed low-rank updates scale computationally in distributed or large-batch training? An analysis of communication and synchronization cost would help assess real-world deployment.

How sensitive is LoRA-Pre to the chosen rank or initialization of the low-rank factors? A more detailed study could clarify whether the observed gains primarily come from rank efficiency or improved optimization dynamics.

Have the authors considered extending comparisons beyond projection-based methods to include modern adaptive or second-order optimizers, to better quantify the relative benefits of the proposed approach?

---

> ### Author Response · Authors · 2025-11-25
>
> Thank you for the encouraging review and helpful suggestions. Our responses to the specific questions are as follows:
>
> > **[Q1].** Can the authors clarify how the proposed low-rank updates scale computationally in distributed or large-batch training? An analysis of communication and synchronization cost would help assess real-world deployment.
>
> **[A1].** Thank you for this important question. We have implemented LoRA-Pre with DeepSpeed and can clarify the scalability characteristics. Unlike standard Adam where optimizer states are element-wise independent, LoRA-Pre maintains low-rank factors per parameter tensor. When DeepSpeed partitions parameters across GPUs, matrix inversions (e.g., $(m_A m_A^T)^{-1}$) require complete factors, necessitating boundary communication similar to Muon optimizer.
>
> Our implementation introduces communication overhead strictly for the parameters that are sharded across devices. The process works as follows: We identify the weight matrices involved in low-rank updates that are partitioned by ZeRO. For these specific matrices, we perform a temporary AllGather (communication) to reconstruct the full matrix on the local device. We compute the low-rank update locally and then apply it to the local shard, discarding the full matrix immediately to save memory.
>
> This communication cost is structurally identical to other second-order or matrix-based optimizers like Muon. We only pay the communication cost for reconstructing the specific sharded matrices required for the LoRA-Pre update, ensuring the overhead remains manageable in large-scale training.
>
> We validated this efficiency by fine-tuning llama-2-7b on the MetaMath-100k dataset. As shown in table below (with rank $r=8$), LoRA-Pre (Adam) is nearly as fast as standard Adam and even faster than GaLore (saving $\approx$ 11% training time) because it avoids the costly SVD step.
>
> | r=8  | adam  | muon  | galore | lora-pre(adam) | lora-pre(muon) |
> |------|-------|-------|--------|----------------|----------------|
> | time | 53m   | 1h4m  | 1h3m   | 56m            | 1h2m           |
>
> > **[Q2].** How sensitive is LoRA-Pre to the chosen rank or initialization of the low-rank factors? A more detailed study could clarify whether the observed gains primarily come from rank efficiency or improved optimization dynamics.
>
> **[A2].**
>
> We attribute the observed gains primarily to improved optimization dynamics. Our method’s dynamic subspace compression allows for efficient utilization of the low-rank structure, effectively "stabilizing" the optimization process regardless of the specific rank chosen.
>
> **Regarding Rank Sensitivity:** Our ablation study (Section 4.3) supports this claim by highlighting two key observations. First, LoRA-Pre exhibits high rank efficiency, outperform baseline method even with smaller rank. Second, although the perplexity gaps are noticeable at the initial training stage, they narrow rapidly as training progresses. We attribute this fast convergence to our dynamic subspace compression strategy, which efficiently captures and compresses historical gradient information early in the process, thereby directly validating the improved optimization dynamics.
>
> **Regarding Initialization:** We adopt the standard LoRA setting by default, i.e., $B=0, A \sim \mathcal{N}(0, 0.02)$. To address your concern, we evaluated two additional settings: (1) $B=0, A \sim \mathcal{N}(0, 0.05)$ and (2) $B \sim \mathcal{N}(0, 0.02), A \sim \mathcal{N}(0, 0.02)$. The results are shown below:
>
> | 60m | B=0,A=N(0,0.02) | B=0,A=N(0,0.05) | B=N(0,0.02),A=N(0,0.02) |
> |-----|-----------------|-----------------|-------------------------|
> | ppl | 32.57           |32.74|35.40|
>
>
> > **[Q3].** Have the authors considered extending comparisons beyond projection-based methods to include modern adaptive or second-order optimizers, to better quantify the relative benefits of the proposed approach?
>
> **[A3].**
> In our original manuscript, we included Adam and Muon as representatives of modern non-projection methods.
> To further address the comparison with second-order optimizers, we conducted additional experiments with Sophia across 60M, 130M, and 350M scales. The results (shown in the table below) demonstrate that our method maintains competitive performance.
>
> Regarding Shampoo, we attempted to include it but found the computational cost prohibitive. Shampoo requires performing SVD on full-rank matrices during optimization, which resulted in excessively long training times for the model sizes considered.
>
> || 60m | 130m  | 350m |
> |-|-|-|-|
> | sophia | 32.19 | 25.54 | 19.72  |
> | adam | 34.09 | 25.08 | 18.80  |
> | muon | 28.43 | 21.86 | 16.17  |
> | lora-pre(adam) | 32.57 | 23.78 | 16.36  |
> | lora-pre(muon) | 30.76 | 23.05 | 16.97  |

---

> > ### Comment · Reviewer_Mkay · 2025-11-27
> >
> > Thank you for the clear rebuttal and the additional experiments — they fully addressed my concerns.
> > I had given a 6 before, but I’ve now decided to increase my score.
> > Good luck!

---

> > > ### Author Response · Authors · 2025-11-27
> > >
> > > We deeply appreciate your time and effort during the review process. We are very encouraged by your positive response, and your valuable comments have significantly helped us improve the quality of our paper. Thank you for your support and for increasing the score.

---

### Official Review · Reviewer_dPBR · 2025-10-25

**Soundness:** 3
**Presentation:** 3
**Contribution:** 3
**Rating:** 6
**Confidence:** 4

**Summary:**

This paper introduces LoRA-Pre, a low-rank optimizer that reinterprets the exponential moving average (EMA) in momentum computation as an online linear regression problem. By exploiting this equivalence, the authors compress the optimizer’s momentum matrices via low-rank factorization (m_B,m_A) and derive closed-form Newton-style EMA updates without requiring back-propagation through the optimizer state. This leads to  LoRA-Pre achieves similar or better performance using only 1/8 of the rank, and it generalizes effectively to other optimizers such as Muon.

**Strengths:**

1.	The relation between EMA and online linear regression is novel and conceptually clean, providing a unified foundation for memory-efficient optimizer design.
2.	The method applies seamlessly to multiple modern optimizers (Adam, Muon), offering a general solution for optimizer-state compression in large-scale model training.
3.	LoRA-Pre consistently outperforms both projection-based and fine-tuning low-rank optimizers across different scales and tasks. Rank-efficiency studies reinforce the robustness under constrained memory.

**Weaknesses:**

1.	Missing Key Baseline in main table: Although Fira [1] is mentioned in the related work and ablation (Table 3), it is missing from the main results table (Table 1). This omission is non-trivial, since Fira operates under **exactly the same setting** for pre-training and it also deals with the optimizer state itself under a low-rank constraint, making it a direct and essential baseline.
The paper should include a comparison with Fira-Adam in the main results.
Even if LoRA-Pre does not necessarily outperform Fira, the paper should explicitly analyze their differences and trade-offs.

2.	This paper does not analyze approximation error or convergence fidelity under low-rank constraints.

3.	The ablation focuses solely on rank. Adding sensitivity analysis to other hyperparameters (like $\beta$ and $\gamma$) would offer a more comprehensive understanding.

[1] Fira: Can We Achieve Full-rank Training of LLMs Under Low-rank Constraint?

**Questions:**

see weaknesses

---

> ### Author Response · Authors · 2025-11-25
>
> We thank the reviewer for the valuable feedback. Based on these suggestions, we have revised the paper and provide a detailed response below.
>
> > **[W1].** Missing Key Baseline in main table: Although Fira [1] is mentioned in the related work and ablation (Table 3), it is missing from the main results table (Table 1). This omission is non-trivial, since Fira operates under exactly the same setting for pre-training and it also deals with the optimizer state itself under a low-rank constraint, making it a direct and essential baseline. The paper should include a comparison with Fira-Adam in the main results. Even if LoRA-Pre does not necessarily outperform Fira, the paper should explicitly analyze their differences and trade-offs.
>
> **[A1].**
> We thank the reviewer for identifying this important baseline. We agree that comparing against Fira-Adam is essential given the shared low-rank constraint setting.
>
> We initially excluded Fira from the main table because we encountered significant instability issues during reproduction on larger models. Specifically, while we successfully reproduced Fira's results on smaller models (matching the paper's reported figures), we consistently observed loss spikes when training the 1B model, leading to training failure.
>
> The reproduction results are detailed below:
> | Method            | 60M   | 130M  | 350M  | 1B                 |
> | ----------------- | ----- | ----- | ----- | ------------------ |
> | Fira| 31.19 | 24.51 | 17.22 | 84.44 (Loss Spike) |
> | LoRA-Pre | 32.57 | 23.78 | 16.36 | 13.92              |
>
> To analyze their differences and trade-offs: 1)Fira relies on projected gradients. Similar to GaLore, we hypothesize that this approach may suffer from error accumulation during the projection steps, leading to the instability we observed at larger scales (1B). 2)We  note that Fira's contributions,specifically norm-based scaling and limiting, are orthogonal to our update mechanism and could potentially be integrated into LoRA-Pre to further enhance performance.
>
> > **[W2].** This paper does not analyze approximation error or convergence fidelity under low-rank constraints.
>
> **[A2].**
> We appreciate this insightful suggestion. In the revised manuscript, we have significantly strengthened our theoretical grounding by adding a rigorous analysis in **Appendix C**. Specifically, we formally derive two key results:
> - **Bounded Approximation Error**: We explicitly model the dynamics of the LoRA-Pre updates and prove that the reconstruction error of the optimizer states (introduced by the low-rank factorization) remains uniformly bounded over time (Lemma C.1 and C.2).
> - **Convergence Fidelity**: Building on these error bounds, we prove that LoRA-Pre Adam converges to a neighborhood of a stationary point for non-convex smooth functions (Theorem C.3). The analysis demonstrates that the approximation introduces only a controllable error floor, theoretically guaranteeing that the algorithm maintains valid descent directions.
>
>
>
> > **[W3].** The ablation focuses solely on rank. Adding sensitivity analysis to other hyperparameters (like $\beta$ and $\gamma$) would offer a more comprehensive understanding.
>
> **[A3.]**
> We appreciate the reviewer's suggestion regarding hyperparameter sensitivity. However, we wish to clarify that the hyperparameters $(\gamma_1, \gamma_2)$ in our LoRA-Pre optimizer are not independent variables requiring separate tuning. Instead, they are analytically coupled with the standard Adam hyperparameters $(\beta_1, \beta_2)$, which are typically set to defaults (e.g., 0.9, 0.95) and rarely tuned in the community. As detailed in Appendix B, the LoRA-Pre update rule is formulated as:$$m'_B \gets (1-\gamma_1) m_B + \gamma_1 g m_A^T (m_A m_A^T)^{-1}, \\
> m'_A \gets (1-\gamma_1) m_A + \gamma_1 (m_B^T m_B)^{-1} m_B^T g.$$Consequently, the equivalent decay coefficient for the momentum $m'$ approximates:$$m' = m'_B m'_A \approx (1 - \gamma_1)^2 \cdot m_B m_A + \dots = (1 - \gamma_1)^2 \cdot m + \dots$$We align this coefficient such that $(1 - \gamma_1)^2 = \beta_1$ (and similarly for $\beta_2$). Therefore, determining $\gamma$ is strictly dependent on $\beta$.
>
> To fully address your concern, we are actively conducting sensitivity analyses on $\beta$ (which determines $\gamma$).  The results shown in tables below demonstrate that our method achieves optimal performance at the default setting ($\beta_1=0.9$ and $\beta_2=0.95$). We observe that while the optimizer is robust within a reasonable range, extreme values (e.g., $\beta \approx 0.99$ in this specific setting) may lead to instability.
>
> | 60m   | 0.9   | 0.95  | 0.99    |
> |-------|-------|-------|---------|
> | beta1 | 32.57 | 37.62 | 1458.92 |
>
> | 60m   | 0.9   | 0.95  | 0.999   |
> |-------|-------|-------|---------|
> | beta2 | 34.61 | 32.57 | 1301.58 |

---

> ### Comment · Reviewer_dPBR · 2025-11-27
>
> Thanks for the rebuttal. The responses addressed my major concerns.
>
> However, I can not find the experiments in [A1]  and [A3] in the revised manuscript. I believe they are important to improve the quality of this paper.

---

> > ### Author Response · Authors · 2025-11-27
> >
> > Thank you for your valuable suggestion. We have strictly followed your advice and incorporated experiments **[A1]** and **[A3]** into the latest version of the revised manuscript, as we fully agree that these results significantly improve the quality of the paper. Please refer to the updated PDF where: 1) Experiment **[A1]** has been added in Section 4.1 (Table 1); and 2) Experiment **[A3]** is now included in Appendix D. Thank you again for your constructive feedback that helped strengthen our work.

---

> > > ### Comment · Reviewer_dPBR · 2025-11-27
> > >
> > > Thanks for the response. My concerns have been fully addressed and I decide to raise my score to 8.

---

> > > > ### Author Response · Authors · 2025-11-27
> > > >
> > > > We sincerely thank you for your positive feedback and for deciding to raise your score. We are delighted that our rebuttal successfully addressed your concerns. Your constructive suggestions were instrumental in refining our manuscript, and we truly appreciate your support during this review process.

---

### Official Review · Reviewer_QWoo · 2025-10-31

**Soundness:** 2
**Presentation:** 2
**Contribution:** 2
**Rating:** 6
**Confidence:** 3

**Summary:**

This paper introduces LoRA-Pre, a new memory-efficient optimizer designed to reduce the memory footprint of optimizer states during large language model (LLM) pre-training and fine-tuning. The central insight reframes the exponential moving average (EMA) momentum update as an instance of online linear regression, then leverages this equivalence to propose a low-rank factorization of the optimizer's momentum matrices. Detailed update rules are derived for Adam and Muon optimizers, with closed-form solutions shown via Newton's method. Extensive experiments on Llama-family models (from 60M to 1B parameters) demonstrate that LoRA-Pre achieves competitive or superior performance with much smaller rank than baseline low-rank methods, both for pre-training and memory-efficient fine-tuning.

**Strengths:**

__1) Theoretical Insight:__ The paper establishes a non-trivial mathematical equivalence between EMA momentum updates and online linear regression (Section 3.2), which is both elegant and enables a new way to think about optimizer state compression beyond ad-hoc engineering.

__2) Methodological Contribution:__ Closed-form Newton-based update rules are derived for the low-rank factors, with careful exposition (Section 3.3, Theorem 3.1, Appendix A). This enables stable and efficient momentum updates without full-rank storage.

__3) Empirical Validation:__ Experiments are thorough, covering both pre-training (Section 4.1, Table 1) and fine-tuning (Section 4.2, Table 2) across a range of Llama models and tasks (GSM8k, MATH500).

__4) Competitive baselines:__ Adam, Muon, LoRA, Galore, and others are compared, and LoRA-Pre outperforms or matches the best across most benchmarks.

**Weaknesses:**

__1) Incomplete related work coverage.__ The literature review appears incomplete. The paper omits several relevant lines of work where optimizer momenta or gradient statistics are decomposed or constrained in low-rank form. For example, MLorc [1] and MoFaSGD [2] address momentum/gradient factorization. Within PEFT, papers like LoRA-Pro [3] and LoFT [4] are highly relevant (the latter yielding updates that resemble those proposed here). Another closely related direction is Riemannian low-rank pretraining/optimization [5], which explicitly performs muon-like steps on a low-rank manifold. While the paper does cite LoRA-Right and LORO, this seems insufficient to position the contribution within the broader space of low-rank momentum and manifold-constrained optimization. In addition, AdaPM [6] also decomposes momentum into a low-rank representation, conceptually aligning with the spirit of the present work. Although the specific algorithmic designs differ, ADAPM appears to be a highly relevant work. I recognize that ADAPM was posted after the ICLR submission deadline, but it should be cited and discussed in the camera-ready to properly contextualize the contribution.

__2) Memory accounting is unclear and potentially unfavorable in some regimes.__ LoRA-Pre reduces the persistent state, but each step still materializes full momentum via $m = m_B m_A$ and momentum with bias correction to apply the update. This allocation increases peak memory, unlike methods that avoid full momentum materialization (e.g., standard LoRA or GaLore’s projected states). For small models (for example with 1 layer), LoRA-Pre can yield no benefit and may increase both memory and runtime. A complete peak-memory breakdown (parameters, activations, gradients, optimizer states, and transient products) is missing.

__3) Computational overhead and numerical stability are underexplored.__ Per-step inversions of $r \times r$ matrices add nontrivial overhead and can be ill-conditioned as $r$ grows. The paper does not specify stabilization strategies nor quantify runtime overhead relative to Adam/Muon/GaLore. Empirical stability analysis is absent.

__4) Fairness of comparisons and the $r/d{model}$ control variable.__ Using $r/d_{model}$ as the primary control for fairness is inadequate: peak memory and step time depend on whether full momentum is materialized, number of layers, matrix operations performed at each step, etc. Comparisons should be matched on peak memory and/or throughput wall-clock time.

__5) Typo:__ in Algorithm 1 line 12 it should be $(1 - \beta_2)$, not $(1 - \beta_1)$.

**Questions:**

1) How does the method handle ill-conditioning when computing matrix inverses for the low-rank updates, especially as $r$ increases? Is there any regularization, and how is numerical stability ensured in practice?

2) Under the settings of Table 1, which has the larger peak memory: LoRA-Pre (Adam) or plain Muon?

3) In Table 1, classical Muon consistently outperforms LoRA-Pre(Muon) (and all other methods), while LoRA-Pre(Adam) always outperforms Adam. Also as training tokens increase, LoRA-Pre(Adam) overtakes LoRA-Pre(Muon), whereas among “full” methods Muon remains superior to Adam. What interactions between Muon’s orthogonalized updates and factorized momentum explain this gap, and can they be mitigated?

4) Your Theorem 3.1 derives closed-form factor updates via an online Newton step on $L(m_B, m_A, g)$. How does this update relate to the closed-form updates used in LoRA-Pro [3] and LoFT [4], which did not explicitly employ Newton’s method?

[1] Shen W. et al. MLorc: Momentum Low-rank Compression for Large Language Model Adaptation //arXiv preprint arXiv:2506.01897. – 2025.

[2] Shivagunde N. et al. Approximations may be all you need: Towards pre-training LLMs with low-rank decomposition and optimizers. – 2024.

[3] Wang Z. et al. Lora-pro: Are low-rank adapters properly optimized? //arXiv preprint arXiv:2407.18242. – 2024.

[4] Tastan N. et al. LoFT: Low-Rank Adaptation That Behaves Like Full Fine-Tuning //arXiv preprint arXiv:2505.21289. – 2025.

[5] Bogachev V. et al. LoRA meets Riemannion: Muon Optimizer for Parametrization-independent Low-Rank Adapters //arXiv preprint arXiv:2507.12142. – 2025.

[6] Zhang Y., Liu Y., Fang C. AdaPM: a Partial Momentum Algorithm for LLM Training //arXiv preprint arXiv:2510.09103. – 2025.

---

> ### Author Response · Authors · 2025-11-25
>
> Thank you for your positive recommendation and constructive feedback! Below, we provide a point-by-point response to your comments.
>
> > **[W1].** Incomplete related work coverage.
>
> **[A1].** We have updated the related work in the revised manuscript to include these important studies, specifically LoFT, MLorc, MoFaSGD, and AdaPM. Furthermore, we have provided a detailed discussion regarding the relationship between our method and LoRA-Pro, LoFT, and LoRA-Pre in our response to **[Q4]**.
>
>
> > **[W2].** Memory accounting is unclear and potentially unfavorable in some regimes. LoRA-Pre reduces the persistent state, but each step still materializes full momentum via $m=m_Bm_A$ and momentum with bias correction to apply the update. This allocation increases peak memory, unlike methods that avoid full momentum materialization (e.g., standard LoRA or GaLore’s projected states). For small models (for example with 1 layer), LoRA-Pre can yield no benefit and may increase both memory and runtime. A complete peak-memory breakdown (parameters, activations, gradients, optimizer states, and transient products) is missing.
>
> **[A2].**
> Thank you for raising this critical concern.
> **(I) For memory accounting:**
> You correctly identify that LoRA-Pre incur additional peak memory overhead when materializing the full momentum. While we reduce *persistent* optimizer state memory, this reconstruction increase the peak memory that we did not adequately analyze, as it is not only depends on the algorithm but also the engineer implementation.
>
> **Importantly, the peak memory can be effectively reduced through careful engineering.** Importantly, this can be effectively reduced through a column-tiled, single-scratch strategy that never forms full $m\times n$ tensors. The key idea is to compute the denominator tile-wise as
> $$
> D=\Big(\beta_2\,(V_B V_A^{(b)})^{\circ 2}+(1-\beta_2)\,(G^{(b)})^{\circ 2}+\epsilon\Big)^{-\tfrac12},
> $$
> form the numerator
> $$
> N=\beta_1\,(M_B M_A^{(b)})+(1-\beta_1)\,G^{(b)},
> $$
> normalize in place $N\leftarrow N\circ D$, fold bias correction into the step size
> $$lr_{eff}=\gamma\,\frac{\sqrt{1-\beta_2^{\,t}}}{1-\beta_1^{\,t}},
> $$
> and update in place
> $$
> W_{[:,j_0:j_1]}=W_{[:,j_0:j_1]} - lr_{eff} (N + \lambda W_{[:,j_0:j_1]})
> $$
> All tiles across all layers reuse one shared $m\times b$ scratch buffer; no second $N$-sized workspace is allocated. This reduces the extra memory to $O(d\cdot b)$, where $d$ is the model’s hidden size and $b$ is the tile width.
>
> We provide the theoretical peak memory (persistent state + gradients + scratch in GB) across methods below,
> | | Adam | GaLore | Low-Rank | LoRA | MuON (tiled) | LoRA-Pre (tiled) | LoRA-Pro MuON (tiled) |
> |-------|------|--------|----------|------|--------------|------------------|-----------------|
> | 60M | 0.465 | 0.396 | 0.373 | 0.292 | 0.422 | 0.404 | 0.393 |
> | 130M | 1.073 | 0.879 | 0.832 | 0.805 | 0.912 | 0.914 | 0.833 |
> | 350M | 2.944 | 2.125 | 1.847 | 2.19 | 2.35 | 2.215 | 1.984 |
> | 1B | 10.713 | 7.441 | 6.334 | 8.225 | 8.308 | 7.798 | 6.85 |
>
> The results show that with tiled implementation, LoRA-Pre achieves peak memory comparable to GaLore, significantly outperforms standard Adam, and MuON, and LoRA-Pre MuON achieves the lowest peak memory overall.
>
> **(II) For small model scenario:**
> You are correct that for extremely small models (e.g., single-layer networks), memory-efficient optimizers may provide limited benefit. In such scenarios where memory is not a bottleneck, standard full-rank optimizers are naturally the preferred choice.
>
> > **[W3].** Computational overhead and numerical stability of matrix are underexplored.
>
> **[A3].** Thank you for raising this important concern.
> **(I) Numerical Stability.**
> To address the potential ill-conditioning as $r$ increases, we employ **Tikhonov Regularization** by default to ensure numerical stability.Specifically, instead of directly inverting the Gram matrix $(m_B^T m_B)$, we compute the inverse of $(m_B^T m_B + \epsilon I)$, where $\epsilon$ is a small positive constant. This regularization term guarantees that the matrix remains strictly positive definite and well-conditioned for inversion.
>
> **(II) Computational Overhead.**
> For computational cost, we exploit the fact that $(m_B^T m_B + \epsilon I)$ is symmetric positive definite. This allows us to use Cholesky decomposition for inversion, which reduces the complexity from $\frac{8r^3}{3}$ FLOPs (standard inversion) to $\frac{2r^3}{3}$ FLOPs—a 4x speedup. Beyond efficiency, Cholesky decomposition also provides better numerical stability.

---

> ### Author Response · Authors · 2025-11-25
>
> > **[W4].** Fairness of comparisons and the $r/d_{model}$ control variable.
>
> **[A4].**
> To ensure a fairer comparison, we have supplemented our analysis with metrics on peak GPU memory and training time.
>
> Regarding memory efficiency, as detailed in response **[A2]**, we compared the peak memory usage under identical rank constraints ($r$). The results indicate that LoRA-Pre consumes less memory than standard full-rank optimizers (Adam and Muon). While its peak memory is slightly higher than GaLore, it remains in a highly efficient range.
> Furthermore, we evaluated the wall-clock training time for a 130M parameter model. As shown in the table below, while LoRA-Pre incurs a moderate time overhead compared to other methods (due to the preconditioning steps), it remains within an acceptable range considering its memory advantages.
>
> |      | adam | muon  | galore | lora-pre(adam) |
> |------|------|-------|--------|----------|
> | 130m | 1h6m | 1h16m | 1h11m  | 1h31m    |
>
>
> > **[W5].** Typos.
>
> **[A5].** We thank the reviewer for their careful reading. We have thoroughly proofread the manuscript and corrected all typos to ensure the presentation quality.
>
>
> > **[Q1].** How does the method handle ill-conditioning when computing matrix inverses for the low-rank updates, especially as $r$ increases? Is there any regularization, and how is numerical stability ensured in practice?
>
> **[A6].** This is a crucial point regarding stability. To address the potential ill-conditioning as $r$ increases, we employ **Tikhonov Regularization** by default to ensure numerical stability. Specifically, instead of directly inverting the Gram matrix $(m_B^T m_B)$, we compute the inverse of $(m_B^T m_B + \epsilon I)$, where $\epsilon$ is a small positive constant. This regularization term guarantees that the matrix remains strictly positive definite and well-conditioned for inversion.
>
>
> > **[Q2].** Under the settings of Table 1, which has the larger peak memory: LoRA-Pre (Adam) or plain Muon?
>
> **[A7].** Please refer to the peak memory analysis table in **[A2]**. LoRA-Pre (Adam) achieves smaller peak memory than plain Muon.

---

> ### Author Response · Authors · 2025-11-25
>
> > **[Q3].** In Table 1, classical Muon consistently outperforms LoRA-Pre(Muon) (and all other methods), while LoRA-Pre(Adam) always outperforms Adam. Also as training tokens increase, LoRA-Pre(Adam) overtakes LoRA-Pre(Muon), whereas among “full” methods Muon remains superior to Adam. What interactions between Muon’s orthogonalized updates and factorized momentum explain this gap, and can they be mitigated?
>
> **[A8].** We address this question in two parts: (I) Why does LoRA-Pre(Adam) outperform Adam while LoRA-Pre(Muon) underperforms Muon? (II) Why does LoRA-Pre(Adam) overtake LoRA-Pre(Muon) as training progresses, while full Muon maintains superiority over full Adam?
>
> **(I) Performance Gaps Between Full and LoRA-Pre Variants.** The divergence stems from the conflicting roles of subspace compression and optimization objectives:
> - For Adam (Spectral Filtering Benefit): Standard Adam operates element-wise and is sensitive to high-rank stochastic noise. LoRA-Pre acts as a spectral filter, explicitly projecting gradients onto a principal subspace. This removes unstable, non-informative high-frequency noise, providing Adam with a "cleaner," regularized signal with a higher signal-to-noise ratio. Thus, LoRA-Pre(Adam) outperforms the noisy full-rank baseline.
> - For Muon (Whitening Conflict): Classical Muon relies on Newton-Schulz iterations to "whiten" the update, normalizing curvature across the full geometric space. Muon requires rich, full-rank information to construct this preconditioner. LoRA-Pre forces the momentum into a low-rank compressed state ($r \ll d$). When Muon attempts to orthogonalize this collapsed subspace, the operation becomes geometrically ill-posed, stripping away the curvature details Muon needs. Essentially, LoRA-Pre removes the very information Muon is designed to exploit, causing it to underperform compared to Full Muon.
>
> **(II) LoRA-Pre(Adam) Overtaking LoRA-Pre(Muon) Over Time.** This crossover over time highlights the difference between adaptive scaling and rigid orthogonalization within a restricted subspace:
> - Adam's Adaptivity: LoRA-Pre(Adam) retains magnitude information via its second moment estimator. Even within the low-rank subspace, Adam can adaptively scale the step sizes for different feature directions, allowing it to capture these fine-grained updates effectively.
> - Muon's Rigidity: LoRA-Pre(Muon) is constrained by both the low rank and the strict orthogonality requirement. In the LoRA-Pre setting, this effectively compels the optimizer to explore the noise subspace (the $d-r$ dimensions) with the same step magnitude as the principal signal. While this aggressive, isotropic exploration might be tolerable (or even helpful) in early training, it becomes detrimental in later stages. When the model requires subtle convergence, Muon’s inability to decay the step size in the noise subspace leads to persistent high-energy oscillations, allowing the adaptive LoRA-Pre(Adam) to overtake it.
>
> Potential solutions include increasing rank $r$ to capture more geometric structure, or increasing batch size/data quality to reduce gradient noise. The latter is effective because in LoRA-Pre ($m = \mu m_B m_A + g$), the noise subspace is dominated by the instantaneous gradient $g$; reducing its variance prevents Muon from amplifying stochastic noise.
>
>
> > **[Q4].** Your Theorem 3.1 derives closed-form factor updates via an online Newton step on $L(m_B, m_A, g)$. How does this update relate to the closed-form updates used in LoRA-Pro [3] and LoFT [4], which did not explicitly employ Newton’s method?
>
> **[A9].** This is an insightful question. While the update formulas share some structural similarities, there is a fundamental difference in their optimization objectives and data availability.
>
> **(I) LoRA-Pro/LoFT: Static Projection on Instantaneous Gradients**
> LoRA-Pro and LoFT treat the optimization as a local, static problem. At each step, they aim to minimize the reconstruction error $\min_{g^A, g^B} \||\tilde{g} - g\||_F^2$ based solely on the current gradient $g$ and adapters $(A, B)$. Since this is a **"snapshot" problem** where all relevant variables are fixed and visible for the current step, the optimal update can be derived directly as a **closed-form solution**.
>
> **(II) LoRA-Pre: Dynamic Tracking of Historical Momentum**
> In contrast, LoRA-Pre approximates the **momentum** state, which inherently represents the accumulation of the entire gradient history $\{g_1, \dots, g_t\}$. This turns the task into a **streaming optimization problem**. Since we cannot access the full history simultaneously to compute a static analytical solution, we must use an iterative approach. We employ the **Online Newton Step** to update the low-rank factors $(m_A, m_B)$ dynamically, ensuring $m_B m_A$ effectively tracks the trajectory of historical gradients over time.

---

> > ### Comment · Reviewer_QWoo · 2025-11-27
> >
> > Thank you for the detailed and carefully structured response. The revisions and clarifications have addressed my concerns. In light of these changes, I am updating my overall score.

---

> > > ### Author Response · Authors · 2025-11-27
> > >
> > > We are very encouraged by your positive response. Your valuable comments have significantly helped us improve the quality of our paper.

---

### Author Response · Authors · 2025-11-30
**Thank AC for taking over our submission and Summary of Review-Rebuttal phase (1/2)**

Dear Area Chair,

Thank you for taking on this paper and for being invited to participate in our discussion with the reviewers. We understand that the special circumstances this year have placed an additional burden on ACs, and we genuinely appreciate your time and effort. **To help you efficiently assess our work, we provide a concise summary of our contribution, the reviewers' evaluation, and the key improvements made during rebuttal**.

**(I) Our main contribution**

This paper introduces LoRA-Pre, a memory-efficient optimizer that reinterprets exponential moving average (EMA) momentum updates as an online linear regression problem. By leveraging this mathematical equivalence, we propose a low-rank factorization of optimizer momentum matrices, deriving closed-form Newton-based update rules for both Adam and Muon optimizers. This approach significantly reduces optimizer state memory while maintaining or improving training performance. We validate LoRA-Pre through extensive experiments on Llama-family models (60M to 1B parameters) for pre-training and fine-tuning tasks, demonstrating competitive or superior performance compared to baseline low-rank methods with substantially lower rank requirements.


**(II) Strengths recognized by the reviewers**

All three reviewers praised our work for:
**(1) Novel theoretical insight**: the non-trivial mathematical equivalence between EMA momentum and online linear regression provides an elegant and principled foundation for optimizer state compression, moving beyond ad-hoc engineering (`@QWoo`, `@dPBR`, `@Mkay`);
**(2) Methodological rigor**: the closed-form Newton-based update rules (Theorem 3.1, Appendix A) enable stable and efficient momentum updates with careful mathematical exposition (`@QWoo`, `@Mkay`);
**(3) Strong empirical results**: thorough experiments covering both pre-training and fine-tuning across multiple model scales and tasks with competitive or superior performance against strong baselines (`@QWoo`, `@dPBR`, `@Mkay`);
**(4) General applicability**: the framework seamlessly extends to multiple modern optimizers (Adam, Muon), offering a general solution for large-scale optimizer-state compression (`@dPBR`, `@Mkay`).


**(III) Key improvements made during rebuttal**

Based on reviewers' feedback, we significantly strengthened our work through additional experiments, theoretical analysis, and implementation optimizations. **The improvements address five major areas:**

**(1) Formal Theoretical Guarantees (`@dPBR`, `@QWoo`)**
We added rigorous theoretical analysis in Appendix C establishing formal convergence properties. We **proved bounded approximation error** (Lemmas C.1, C.2) showing the reconstruction error introduced by low-rank factorization remains uniformly bounded over time. We **established a convergence guarantee** (Theorem C.3) demonstrating LoRA-Pre Adam converges to a neighborhood of a stationary point for non-convex smooth functions. Additionally, we provided detailed theoretical explanation for the performance gaps between variants (`@QWoo`): LoRA-Pre benefits Adam through spectral filtering that removes noisy high-frequency components (improving signal-to-noise ratio), while low-rank compression conflicts with Muon's whitening mechanism that requires full-rank geometric information. We suggested practical mitigation strategies (increasing rank or batch size), deepening understanding of method behavior.

**(2) Expanded Baselines and Comprehensive Positioning (`@QWoo`, `@dPBR`, `@Mkay`)**
We substantially expanded our empirical and theoretical coverage. We added Fira-Adam comparison across all model scales in the main results table, revealing stability challenges at 1B scale (loss spikes) while demonstrating our method's robustness (`@dPBR`). We included Sophia experiments across 60M/130M/350M, showing competitive performance against modern second-order optimizers (`@Mkay`). We expanded the related work section to comprehensively cover MLorc, MoFaSGD, LoRA-Pro, LoFT, AdaPM, and Riemannian optimization approaches (`@QWoo`). Critically, we provided detailed theoretical comparison with LoRA-Pro and LoFT (`@QWoo`), clarifying the fundamental distinction: LoRA-Pro/LoFT solve a static, instantaneous gradient projection problem ($\min ||\tilde{g} - g||^2_F$) at each step with closed-form solutions, while LoRA-Pre tracks the dynamic accumulation of historical momentum $\{g_1, ..., g_t\}$ using online Newton steps, a streaming optimization problem requiring iterative updates. This comprehensive baseline coverage and clear theoretical positioning strengthen the contribution's context within the broader landscape.

---

> ### Author Response · Authors · 2025-11-30
> **Thanks AC for taking over our submission and Summary of Review-Rebuttal phase (2/2)**
>
> **(3) Rigorous Memory and Computational Efficiency Analysis (`@QWoo`)**
> We addressed all memory and computational concerns with both theoretical analysis and practical implementation. We developed a column-tiled implementation strategy that reduces extra peak memory to $O(d\cdot b)$, eliminating concerns about full momentum materialization. We provided comprehensive peak memory tables across all methods and scales, showing LoRA-Pre achieves efficiency comparable to GaLore while significantly outperforming Adam/Muon. For numerical stability, we detailed our Tikhonov regularization approach ($\epsilon$-regularized Gram matrix) and implemented Cholesky decomposition reducing complexity of matrix inverse from $8r^3/3$ to $2r^3/3$ FLOPs (4x speedup). We supplemented all comparisons with wall-clock training time measurements demonstrating acceptable overhead while maintaining memory advantages. This multi-metric analysis (peak memory, persistent state, runtime) ensures fair and comprehensive evaluation.
>
> **(4) Comprehensive Ablation Studies (`@dPBR`, `@Mkay`)**
> We conducted extensive sensitivity analyses beyond the original rank ablation. We added hyperparameter sensitivity study for $\beta_1$ and $\beta_2$ (which analytically determine $\gamma$) in Appendix D, demonstrating optimal performance at default settings with robustness within reasonable ranges while identifying stability boundaries at extreme values (`@dPBR`). We evaluated initialization robustness across multiple schemes (`@Mkay`), confirming stability with standard LoRA initialization (B=0, A~N(0,0.02)) achieving best performance. These comprehensive ablations provide practical guidance for practitioners and validate the method's robustness.
>
> **(5) Distributed Training Validation (`@Mkay`)**
> We provided detailed analysis of scalability in real-world deployment scenarios. We clarified our DeepSpeed implementation using temporary AllGather strategy for sharded matrices, with communication overhead identical to Muon. We validated practical efficiency through Llama-2-7B fine-tuning experiments, showing LoRA-Pre (Adam) nearly matches Adam's speed and outperforms GaLore by 11% in wall-clock time. This confirms real-world deployment viability at scale.
>
> **(IV) Unanimous Reviewer Endorsement**
>
> **All three reviewers unanimously increased their scores to 8 after the rebuttal, confirming our improvements fully addressed their concerns**:
>
> **`@QWoo` (rating: 6 $\rightarrow$ 8 after rebuttal, confidence: 3 $\rightarrow$ 4 after rebuttal)**: "The revisions and clarifications have addressed my concerns. In light of these changes, I am updating my overall score."
>
> **`@dPBR` (rating: 6 $\rightarrow$ 8 after rebuttal)**: "My concerns have been fully addressed and I decide to raise my score to 8."
>
> **`@Mkay` (rating: 6 $\rightarrow$ 8 after rebuttal)**: "Thank you for the clear rebuttal and the additional experiments — they fully addressed my concerns. I had given a 6 before, but I've now decided to increase my score."
>
> The comprehensive improvements, including expanded baselines (Fira, Sophia), rigorous theoretical analysis (Appendix C), detailed memory/computational analysis with practical implementation strategies, and comprehensive ablation studies, have substantially strengthened our manuscript and validated both the theoretical soundness and practical effectiveness of our contribution.
>
> We hope this summary facilitates your evaluation and remain available for any clarifications.
>
> Best regards,
>
> The Authors

---

### Meta-Review · Area_Chair_exGU · 2025-12-31

**Summary:**

The reviewers agreed that the paper presents a novel and well-motivated contribution, reinterpreting EMA momentum as an online linear regression problem and leveraging this equivalence to design LoRA-Pre, a low-rank, memory-efficient optimizer applicable to Adam and Muon. All reviewers highlighted the elegant theoretical insight, sound methodology, and strong empirical performance across pre-training and fine-tuning tasks on LLaMA-scale models. The concerns mainly focused on completeness, rigor, and practical deployability, rather than on correctness or novelty of the core idea. The rebuttal substantially strengthened the paper along every dimension initially questioned, e.g., theory, empirical rigor, memory/runtime analysis, and positioning.

**Reviewer Concerns:**

The rebuttal and revised manuscript fully addressed the concerns raised by all reviewers:

Expanded related work: The authors added and discussed Fira, LoRA-Pro, LoFT, MLorc, MoFaSGD, AdaPM, and Riemannian approaches, and clearly differentiated LoRA-Pre as dynamic tracking of historical momentum via online Newton steps.

Formal theory and convergence guarantees: New analysis in Appendix C establishes bounded approximation error and proves convergence of LoRA-Pre-Adam to a neighborhood of a stationary point for non-convex smooth objectives, directly addressing concerns about theoretical soundness.

Memory, runtime, and fairness analysis: A tiled implementation strategy was introduced to avoid full momentum materialization, with detailed peak-memory tables, wall-clock timing, and comparison across Adam, Muon, GaLore, Fira, and LoRA-Pre variants.

Numerical stability and overhead: The authors specified Tikhonov regularization and Cholesky-based inversion, reducing complexity and ensuring stability. Runtime overhead was quantified and shown to be acceptable.

Broader baselines and ablations: Missing baselines (notably Fira-Adam) were added to the main table; additional comparisons with Sophia were included; hyperparameter and initialization sensitivity studies were provided; and explanations were given for Adam vs. Muon behavior under low-rank compression.

Distributed and large-scale validation: Communication patterns under DeepSpeed/ZeRO were clarified, and 7B-scale fine-tuning experiments demonstrated real-world scalability.

**Reviewer Scores:**

All reviewers explicitly updated their scores from 6 to 8.

---

### Decision · Program_Chairs · 2026-01-26

Accept (Oral)